# Polyrotaxane-based supramolecular theranostics

Guocan Yu[1], Zhen Yang[1,2], Xiao Fu[3,4], Bryant C. Yung[1], Jie Yang[5], Zhengwei Mao[6], Li Shao[5], Bin Hua[5], Yijing Liu[1], Fuwu Zhang[1], Quli Fan[2], Sheng Wang[1], Orit Jacobson[1], Albert Jin[3], Changyou Gao[6], Xiaoying Tang[4], Feihe Huang[5] & Xiaoyuan Chen[1]

The development of smart theranostic systems with favourable biocompatibility, high loading efficiency, excellent circulation stability, potent anti-tumour activity, and multimodal diagnostic functionalities is of importance for future clinical application. The premature burst release and poor degradation kinetics indicative of polymer-based nanomedicines remain the major obstacles for clinical translation. Herein we prepare theranostic shell-crosslinked nanoparticles (SCNPs) using a β-cyclodextrin-based polyrotaxane (PDI-PCL-$b$-PEG-RGD⊃β-CD-NH$_2$) to avoid premature drug leakage and achieve precisely controllable release, enhancing the maximum tolerated dose of the supramolecular nanomedicines. cRGDfK and perylene diimide are chosen as the stoppers of PDI-PCL-$b$-PEG-RGD⊃β-CD-NH$_2$, endowing the resultant SCNPs with excellent integrin targeting ability, photothermal effect, and photoacoustic capability. In vivo anti-tumour studies demonstrate that drug-loaded SCNPs completely eliminate the subcutaneous tumours without recurrence after a single-dose injection combining chemotherapy and photothermal therapy. These supramolecular nanomedicines also exhibit excellent anti-tumour performance against orthotopic breast cancer and prevent lung metastasis with negligible systemic toxicity.

---

[1] Laboratory of Molecular Imaging and Nanomedicine, National Institute of Biomedical Imaging and Bioengineering, National Institutes of Health, Bethesda, MD 20892, USA. [2] Key Laboratory for Organic Electronics and Information Displays & Institute of Advanced Materials (IAM), Jiangsu National Synergetic Innovation Centre for Advanced Materials (SICAM), Nanjing University of Posts & Telecommunications, 210023 Nanjing, China. [3] Laboratory of Cellular Imaging and Macromolecular Biophysics, National Institute of Biomedical Imaging and Bioengineering (NIBIB), National Institutes of Health, Bethesda, MD 20892, USA. [4] School of Life Science, Beijing Institute of Technology, 100081 Beijing, China. [5] State Key Laboratory of Chemical Engineering, Centre for Chemistry of High-Performance & Novel Materials, Department of Chemistry, Zhejiang University, 310027 Hangzhou, China. [6] MOE Key Laboratory of Macromolecular Synthesis and Functionalization, Department of Polymer Science and Engineering, Zhejiang University, 310027 Hangzhou, China. Guocan Yu, Zhen Yang and Xiao Fu contributed equally to this work. Correspondence and requests for materials should be addressed to Z.M. (email: zwmao@zju.edu.cn) or to Q.F. (email: iamqlfan@njupt.edu.cn) or to F.H. (email: fhuang@zju.edu.cn) or to X.C. (email: shawn.chen@nih.gov)

Nanotechnology-based drug delivery systems (DDSs) have shown promising results in medical technology, greatly improving the therapeutic performances of many existing drugs and implementing entirely new therapies[1,2]. Among various drug delivery vehicles, nanoparticles (NPs) fabricated from block copolymers are widely employed to enhance the efficacy and mitigate the harmful side effects of drugs, owing to their ability to extend circulation time and promote uptake via the enhanced permeability and retention (EPR) effect[3]. However, one practical concern with NPs is the premature burst release of loaded drug during systemic circulation caused by large dilution volume, which reduces drug concentration at the target site and increases off target exposure[4]. Covalent crosslinking the shell/core of NPs has emerged as one of the most promising strategies to inhibit de-micellization-associated drug leakage[5,6]. However, overly stable NPs are far from optimal for drug delivery, since drug efficacy is significantly diminished, despite reaching the target site. Development of smart DDSs allowing for tailored release profiles, with excellent spatial and temporal dosage control, therefore, remains a daunting task.

Apart from burst release, tumour recurrence is another obstacle impeding the efficacy of both conventional chemotherapeutic and molecularly targeted agents. Comparatively, laser irradiation-assisted photothermal therapy (PTT), a noninvasive and localised therapeutic modality, is capable of converting near-infrared (NIR) light energy into heat, rising the local temperature to rapidly kill cancer cells[7,8]. Although photothermal platforms are efficient in local ablation of tumours, their activity on cancers distal from the primary tumour is lacking, thereby leading to recurrence. The combination of systemic chemotherapy and PTT, termed as thermo-chemotherapy, has demonstrated efficacy in optimising the outcome of cancer treatments attributing to the synergistic therapeutic effect[9–11]. Followed by photothermal ablation of the primary tumour tissue, the preloaded drugs can be released and introduced to a wider range of cancerous cells by heat-induced disruption of the delivery vehicle and enhancement of vascular permeability, effectively inhibiting tumour recurrence and eliminating the need for multiple doses.

Mechanically interlocked molecules (MIMs), especially rotaxanes and polyrotaxanes, have attracted much attention, not only due to their interesting topological features, but also because of their applications in the field of drug delivery and controlled release[12–14]. Considering their sensitive environment responsiveness and dynamic nature arising from non-covalent driving forces, these MIMs have proven to be promising candidates in the biomedical and pharmaceutical fields. Targeting ligands, therapeutic drugs, and diagnostic agents can be incorporated into the theranostic MIMs by modifying the corresponding building blocks separately, avoiding time-consuming and costly covalent syntheses[15,16]. However, most of the reported MIM-based DDSs have focused on their structure, whereas their distinctive properties are rarely exploited. By leveraging their unparalleled topological structures, stimuli responsiveness, and nanoscale properties, intelligent supramolecular nanomedicines can be developed to enable high blood stability/biocompatibility, tailored release, precise imaging, and superb anti-tumour performance, demonstrating promising potential for clinical translation.

Herein we develop supramolecular nanomedicines by employing polyrotaxane as a theranostic platform, where the amphiphilic diblock copolymer acts as the axle and the primary-amino-containing $\beta$-cyclodextrin ($\beta$-CD-NH$_2$) acts as the wheel, driven by the host–guest complexation between $\beta$-CD-NH$_2$ and poly($\epsilon$-caprolactone) (PCL) segment (Fig. 1). Owing to the existence of $\pi$–$\pi$ stacking interactions between the perylene diimide (PDI) stoppers and hydrophobic interactions between the PCL chains, the resultant polyrotaxane self-assembles into core-shell-

structured NPs, which can be utilised to encapsulate hydrophobic anticancer drugs, such as paclitaxel (PTX) and camptothecin (CPT). Premature drug leakage can be effectively inhibited upon formation of shell-crosslinked NPs (SCNPs), which is realised by reaction of N-hydroxysulfosuccinimide (NHS) ester-activated crosslinker (NHS-SS-NHS) with the primary amine on $\beta$-CD-NH$_2$. The introduction of disulfide bonds confer SCNPs sensitivity towards intracellular glutathione (GSH), achieving specific release of the loaded cargoes inside cells. The maximum tolerated dose (MTD) of the supramolecular nanomedicine is significantly increased by taking full advantage of the smart topological structure of the polyrotaxane. The cyclic peptide working as the other stopper of polyrotaxane endows the SCNPs with excellent targeting ability to specifically deliver drugs to cancer cells overexpressing $\alpha_v\beta_3$ integrin receptor. Additionally, the free amines are able to graft diagnostic agents (Cy5.5 dye and DOTA) for fluorescence imaging and positron emission tomography (PET) imaging, respectively. More interestingly, the PDI stopper showing intensive absorption in the NIR region is an ideal photothermal agent and photoacoustic (PA) probe, thus enriching the theranostic functions of this supramolecular nanomedicine. In vivo studies demonstrated excellent anti-tumour performance and antimetastatic effect, benefiting from synergistic combination of PTT and chemotherapy.

## Results

**Synthesis and characterisation of polyrotaxane.** The maximum internal diameter of $\beta$-CD is 5.8 Å; therefore, we chose PDI (~13 Å in width) and cRGDfK (~18 Å in width) as the stoppers. The polyrotaxane (PDI-PCL-b-PEG-RGD⊃$\beta$-CD-NH$_2$) was synthesised in four steps (Fig. 1; Supplementary Figs. 1–9). First, PDI-PCL was obtained through ring-opening polymerisation by using PDI-OH as an initiator, which further reacted with succinic anhydride to afford PDI-PCL-COOH. After esterification reaction between PDI-PCL-COOH and OH-PEG-Mal, the diblock copolymeric axle (PDI-PCL-b-PEG-Mal) was purified by dialyzing against double-distilled water to remove excess amount of OH-PEG-Mal. Polypseudorotaxane inclusion complex (PDI-PCL-b-PEG-Mal⊃$\beta$-CD-NH$_2$) was formed by the host–guest interactions between PDI-PCL-b-PEG-Mal and $\beta$-CD-NH$_2$, which self-assembled into NPs in aqueous solution. The other side of polypseudorotaxane was locked by employing in situ click chemistry between the maleimides on the surface of the NPs and cRGDfK-SH. In order to balance the drug loading efficiency and crosslink density, the average number of wheels on PDI-PCL-b-PEG-RGD⊃$\beta$-CD-NH$_2$ was maintained around seven by controlling the ratio between $\beta$-CD-NH$_2$ and PDI-PCL-b-PEG-Mal in the click reaction (Supplementary Figs. 10, 11 and Supplementary Table 1).

Various characterisations were applied to confirm the successful synthesis of the polyrotaxane. As shown in gel permeation chromatography (GPC) curves (Fig. 2a), the average molecular weight ($M_n$) of PDI-PCL-b-PEG-RGD⊃$\beta$-CD-NH$_2$ was determined to be 16.8 kDa (Supplementary Fig. 11), which is 8.15 kDa higher than that of PDI-PCL-b-PEG-Mal ($M_n = 8.65$ kDa, Supplementary Fig. 9), providing direct evidence for the formation of polyrotaxane containing around seven $\beta$-CD-NH$_2$ groups. On the other hand, co-existence of the peaks related to the protons on PDI-PCL-b-PEG-Mal and $\beta$-CD-NH$_2$ were observed in the $^1$H NMR spectrum (Supplementary Fig. 10), confirming the formation of a mechanically interlocked molecule. Comparing the integration of the signals related to the protons on $\beta$-CD-NH$_2$ with that of the PEG segment, the average number of $\beta$-CD-NH$_2$ in the polyrotaxane was calculated to be 7.3, which was in agreement with that obtained from GPC study

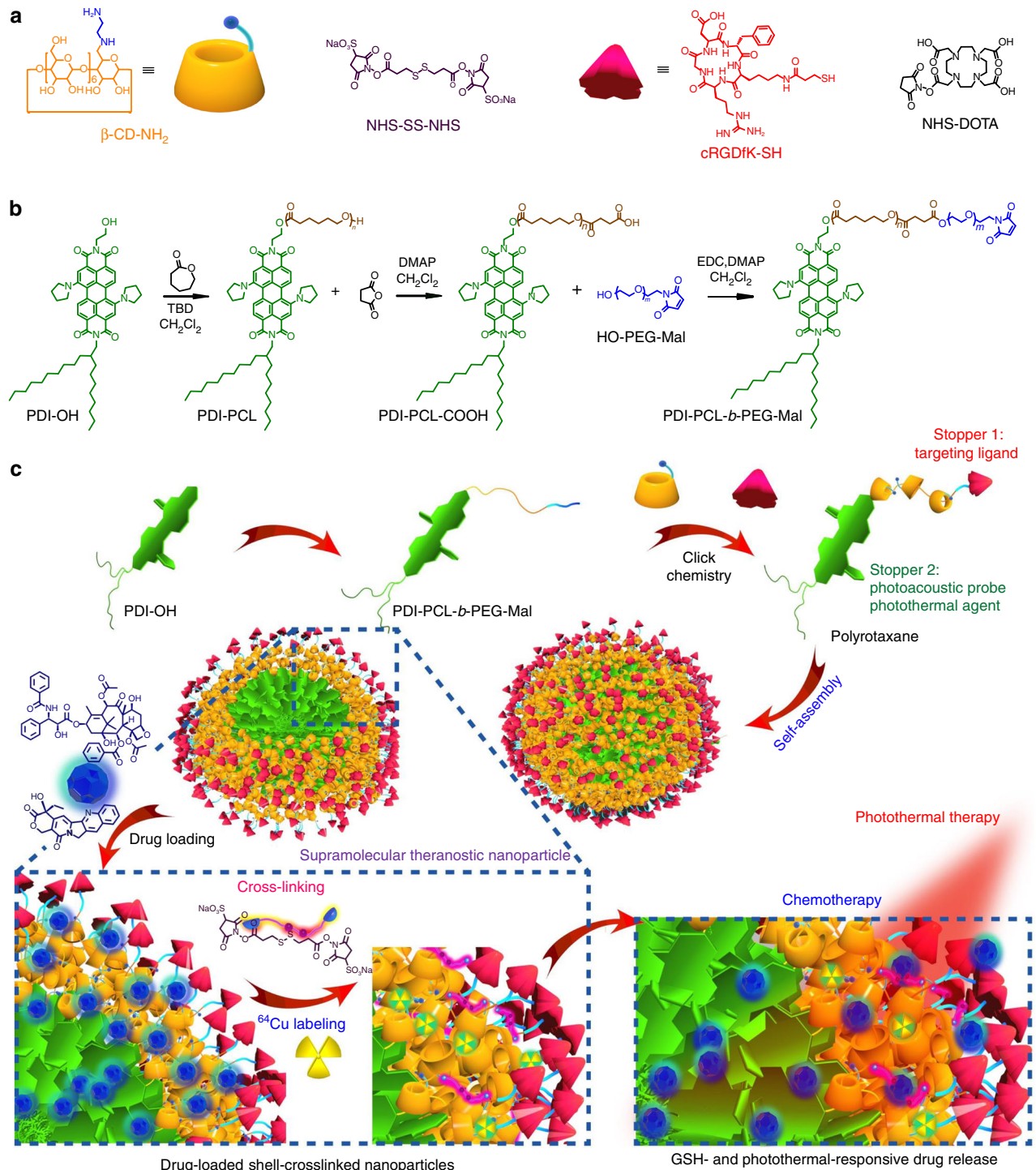

**Fig. 1** Synthesis and fabrication schematics of SCNPs for supramolecular theranostics. **a** Chemical structures and cartoon representations of the building blocks (β-CD-NH₂, NHS-SS-NHS, cRGDfK-SH, and NHS-DOTA). **b** Synthetic route to the polyrotaxane (PDI-PCL-*b*-PEG-RGD⊃β-CD-NH₂). **c** Schematic illustrations of the preparation of drug-loaded SCNPs and dual-responsive drug release

(Supplementary Fig. 10 and Supplementary Fig. 18). In 2D NOESY NMR spectroscopy, strong nuclear overhauser effect (NOE) correlations were found between the peaks corresponding to the protons on β-CD-NH₂ and the central protons of the PCL section (Fig. 2b), suggesting that the hydrophobic PCL chain deeply penetrated into the cavity of β-CD-NH₂.

In the Fourier-transform infrared spectrum of PDI-PCL-*b*-PEG-RGD⊃β-CD-NH₂ (Supplementary Fig. 13), the characteristic peaks for β-CD-NH₂ (1029, 1157, and 3347 cm$^{-1}$,

corresponding to the C–O bending vibration, C–O stretching, and O–H bending vibration, respectively) and PCL segment (1168, 1727, and 2887/2944 cm$^{-1}$, corresponding to symmetric C–O–C stretching, C=O stretching, asymmetric/symmetric –CH₂– stretching, respectively) were observed, indicating the presence of these components. In comparison with the spectrum of PDI-PCL-*b*-PEG-Mal, the intensity of the peak at 1727 cm$^{-1}$ for PDI-PCL-*b*-PEG-RGD⊃β-CD-NH₂ was much lower, because the carbonyl groups of the PCL chains were included in β-CD-

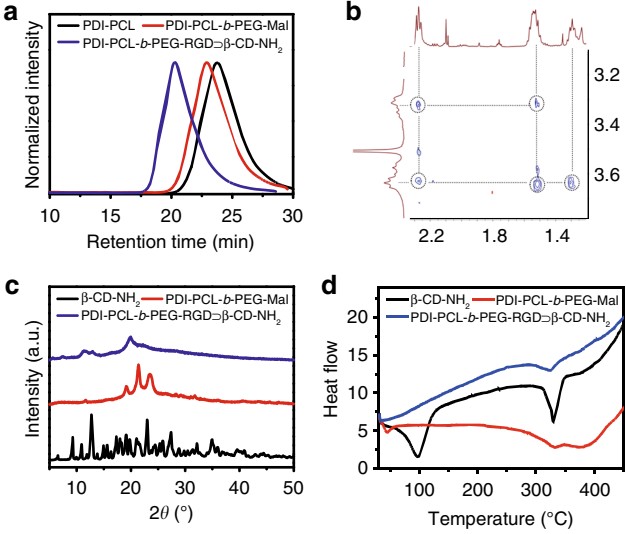

**Fig. 2** Characterisation of polyrotaxane. **a** GPC curves of PDI-PCL, PDI-PCL-*b*-PEG-Mal, and PDI-PCL-*b*-PEG-RGD⊃β-CD-NH₂. **b** 2D NOESY NMR spectrum (500 MHz, 298 K) of PDI-PCL-*b*-PEG-RGD⊃β-CD-NH₂. **c** XRD patterns of β-CD-NH₂, PDI-PCL-*b*-PEG-Mal, and PDI-PCL-*b*-PEG-RGD⊃β-CD-NH₂. **d** DSC thermograms of β-CD-NH₂, PDI-PCL-*b*-PEG-Mal, and PDI-PCL-*b*-PEG-RGD⊃β-CD-NH₂

NH₂ cavities[17]. The X-ray diffraction pattern (XRD) of the freeze-dried PDI-PCL-*b*-PEG-RGD⊃β-CD-NH₂ was substantially different from those of PDI-PCL-*b*-PEG-Mal and β-CD-NH₂ (Fig. 2c). XRD diffractograms of β-CD-NH₂ indicated a series of peaks ranging from 5° to 40°, confirming the macrocyclic structure[17]. PDI-PCL-*b*-PEG-Mal, a semicrystalline material with orthorhombic crystal structure, exhibited two distinct diffraction peaks at 22° and 24°, related to (110) and (200) lattice structures, respectively[18]. For PDI-PCL-*b*-PEG-RGD⊃β-CD-NH₂, the most intense diffraction peak appeared at 19.8°, and the characteristic peaks of PCL broadened and the β-CD-NH₂ peaks diminished upon the formation of polyrotaxane. Differential scanning calorimetry (DSC) indicated that the endothermic peak of PDI-PCL-*b*-PEG-Mal at 41.6°C disappeared upon complexation with β-CD-NH₂ (Fig. 2d), because the crystallisation of PDI-PCL-*b*-PEG-Mal was greatly inhibited by the formation of polyrotaxane, which was consistent with the data obtained from XRD. Moreover, thermogravimetric analyses (Supplementary Fig. 14) and critical aggregation concentration measurements (Supplementary Fig. 20) were conducted to confirm the successful preparation of PDI-PCL-*b*-PEG-RGD⊃β-CD-NH₂ through host−guest chemistry.

**Preparation of the SCNPs as dual-responsive drug delivery vehicles.** Due to its amphiphilicity, PDI-PCL-*b*-PEG-RGD⊃β-CD-NH₂ self-assembled into NPs ranging from 50 to 100 nm in diameter (Supplementary Fig. 21a). The size obtained from transmission electron microscopy (TEM) image was in good agreement with the DLS measurement, which gave an average diameter of 67.1 nm (Fig. 3b). In these NPs, β-CD-NH₂ acting as "supramolecular gates" located at the interface between the hydrophobic core and hydrophilic shell. NHS-SS-NHS containing a disulfide bond was chosen as the crosslinker to prepare SCNPs, endowing them with excellent redox sensitivity. In situ crosslinking was conducted by introducing NHS-SS-NHS into a solution of the self-assembled PDI-PCL-*b*-PEG-RGD⊃β-CD-NH₂ NPs. Consumption of amine groups on β-CD-NH₂ during -NHS/NH₂- coupling reaction resulted in a reduction of zeta potential (Fig. 3c), verifying the formation of shell-crosslinked

structures. No obvious changes in the morphology and diameter of the assemblies before and after crosslinking demonstrated that the crosslinking took place predominantly within individual assemblies (Fig. 3a, b). Due to the difference in extracellular (~2–10 μM) and intracellular (~2–10 mM) GSH concentrations[19], the structure of SCNPs were maintained in the delivery process, while intracellular burst release of drug was achieved through the cleavage of disulfide bonds by high concentration of GSH. Furthermore, since the intracellular GSH concentration in cancerous cells is much higher than that in the normal cells[19], this differentiating feature is expected to reduce non-specific release in normal cells, thus decreasing the side effects of drug-loaded SCNPs.

Acting as a stopper of polyrotaxane, the PDI group showed high absorbance in the NIR region between 600 and 750 nm (Supplementary Fig. 22), suggesting its potential as a photothermal agent. As shown in Fig. 3d, e, the solution containing SCNPs exhibited concentration power-dependent and laser power-dependent temperature increase. For example, the solution temperature increased from 25.4 to 61.9 °C within 150 s upon laser irradiation (0.5 W cm⁻²) when the concentration of SCNPs was kept at 0.5 mg mL⁻¹ (Supplementary Fig. 23). The photothermal conversion efficiency was determined to be 0.36 according to the energy balance on the system as determined by the model reported previously[20]. More interestingly, the photostability of SCNPs was excellent, with less than 5% reduction in absorption observed after five cycles of irradiation at 0.5 W cm⁻² (Fig. 3f). Comparatively, nearly 90% absorption was annihilated for the commercially used indocyanine green (ICG) dye under the same laser density. Notably, severe photobleaching was observed for ICG aqueous solution, with the colour turning from cyan to nearly transparent, while the SCNP solution retained the same colour after five cycles of irradiation (Fig. 3g). During the five cycles of laser irradiation, SCNPs maintained constant photothermal conversion efficiency, whereas ICG gradually lost its photothermal capacity (Fig. 3h).

The PTX-loaded SCNPs (SCNPs@PTX) were prepared through a co-assembly technique with a drug loading efficiency of 35.4%. Compared with the blank SCNPs, negligible changes in morphology were detected upon formation of SCNPs@PTX (Supplementary Fig. 21b). However, a slight increase in diameter was observed by DLS measurement, with the average diameter increased to 128 nm (Supplementary Fig. 26). Importantly, SCNPs@PTX were stable in PBS containing 10% FBS (Supplementary Fig. 27), confirming the high colloidal stability of SCNPs@PTX in biological buffer. Compared with the amphiphilic axle, the polyrotaxane exhibited higher drug loading capability and stability of the resultant SCNPs@PTX (Table 1). For PDI-PCL-*b*-PEG-Mal, the PCL chains partially crystallised due to the tight arrangement of the hydrophobic segments, which greatly decreased the drug loading capability and the resultant stability of the drug-loaded NPs (Fig. 4a). For PDI-PCL-*b*-PEG-RGD⊃β-CD-NH₂, the crystallisation of the polymeric chains was remarkably inhibited by introducing β-CD-NH₂ attributing to their possible shuttle motion along the axle. This unique topological structure made the SCNPs behave like "nanosponges" to stably encapsulate large amount of drugs, in sharp contrast with traditional polymeric DDSs (Fig. 4b).

The drug release profiles were measured in the absence and presence of GSH with or without laser irradiation over a period of 24 h (Fig. 3i). Compared with the non-crosslinked NPs showing 61.6% PTX release over 24 h, shell crosslinking created an additional barrier that hindered the drug from leaking through the shell, thus reducing the release rate (6.27% within 24 h). An accelerated drug release profile was observed for SCNPs@PTX in the presence of 10.0 mM GSH, as the "supramolecular gates" of

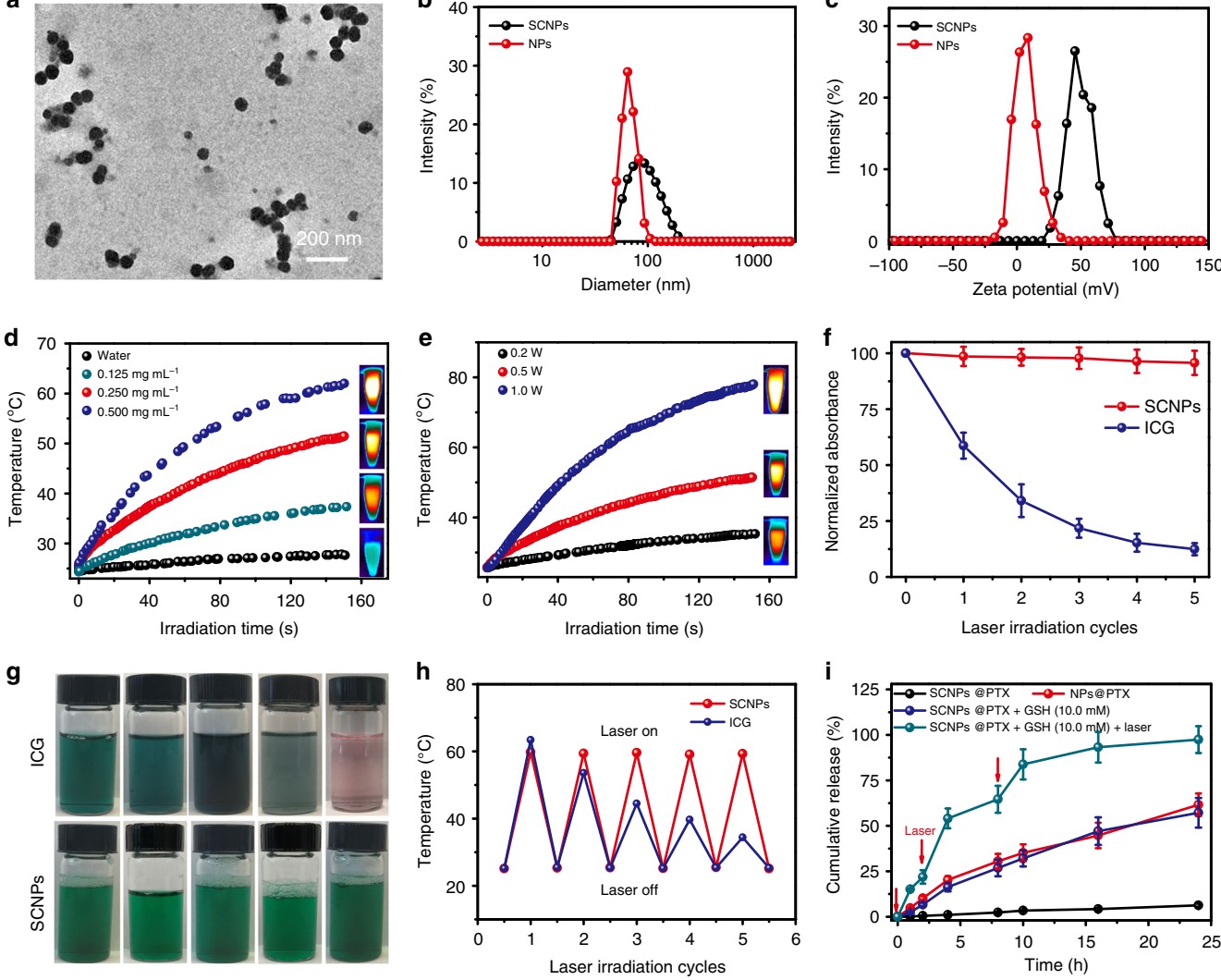

**Fig. 3** Preparation of SCNPs and stimuli-responsive drug release. **a** TEM image of SCNPs. **b** DLS results and **c** zeta potential of the NPs formed from PDI-PCL-*b*-PEG-RGD⊃β-CD-NH$_2$ and SCNPs. **d** The photothermal curves of pure water and SCNPs at different concentrations under 671 nm laser irradiation at a power density of 0.5 W cm$^{-2}$. **e** The photothermal curves of SCNPs (0.250 mg mL$^{-1}$) under 671 nm laser irradiation at different power densities. **f** The changes in absorbance intensity, **g** photographs, and **h** thermal curves of SCNPs and ICG after five cycles of irradiation, respectively. The test laser wavelengths of SCNPs and ICG were 671 and 780 nm, respectively, with a power density of 0.5 W cm$^{-2}$. **i** Controlled release profiles of SCNPs@PTX and NPs@PTX under different conditions. Data are expressed as means ± s.e.m. ($n = 3$)

**Table 1 Drug loading capability and stability evaluations**

|  | Polymer (mg) | PTX (mg) | DLE (%)[a] | DLC (%)[b] | Size (nm)[c] | Stability |
|---|---|---|---|---|---|---|
| Polyrotaxane | 100 | 20 | 98.1 | 16.4 | 92.3 ± 10.2 | No precipitation over 2 d |
|  | 100 | 30 | 98.6 | 22.8 | 105 ± 14.7 | No precipitation over 2 d |
|  | 100 | 40 | 96.5 | 27.8 | 119 ± 12.6 | No precipitation over 2 d |
|  | 100 | 50 | 95.3 | 32.3 | 113 ± 17.8 | No precipitation over 2 d |
|  | 100 | 60 | 91.3 | 35.4 | 128 ± 15.2 | No precipitation over 2 d |
|  | 100 | 70 | 76.4 | 34.8 | 231 ± 30.7 | Precipitation within 8 h |
| PDI-PCL-*b*-PEG-Mal | 100 | 20 | 90.4 | 15.3 | 85.1 ± 13.1 | No precipitation over 2 d |
|  | 100 | 30 | 86.3 | 20.6 | 142 ± 18.5 | No precipitation over 2 d |
|  | 100 | 40 | 74.2 | 22.9 | 212 ± 38.7 | Precipitation within 6 h |
|  | 100 | 50 | 67.7 | 25.3 | 357 ± 54.2 | Precipitation within 2 h |
|  | 100 | 60 | 54.6 | 24.7 | 547 ± 84.7 | Precipitation within 2 h |

[a]Drug loading efficiency (DLE) = $m_{load}/m_{add}$ * 100%, where $m_{add}$ and $m_{load}$ represent the drug mass added during the preparation of drug-loaded NPs and loaded by the NPs, respectively
[b]Drug loading content (DLC) = $m_{load}/(m_{load} + m)$ * 100%, where $m$ represents the polymer mass used during the preparation of drug-loaded NPs
[c]The size of the drug-loaded NPs was measured by DLS

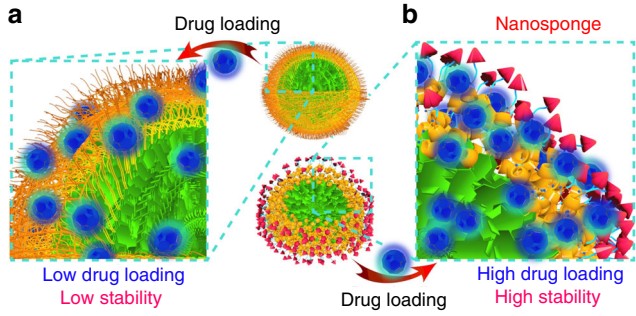

**Fig. 4** Schematic illustration of drug encapsulation. Schematic comparison of drug loading efficiency and stability between **a** NPs and **b** SCNPs formed from PDI-PCL-*b*-PEG-Mal and polyrotaxane, respectively

SCNPs were opened by the cleavage of disulfide bonds. Furthermore, photothermal effect facilitated the escape of loaded PTX from SCNPs in the presence of 10.0 mM GSH by increasing the solution temperature (Fig. 3i). NHS-CC-NHS was chosen as a non-degradable crosslinker to prepare SCNPsCC as a control. Only 19.5% of the loaded PTX released in the presence of 10.0 mM GSH upon irradiation (Supplementary Fig. 28). Because the gates were kept closed, the drug diffused slowly even as solution temperature increased. From these studies, we concluded that premature release was effectively avoided and the dual-responsive release was achieved through supramolecular strategy.

**In vitro targeted drug delivery and thermo-chemotherapy.** Interestingly, cRGDfK acting as the stopper of polyrotaxane works as an excellent targeting ligand to specifically deliver the drug-loaded SCNPs to cancer cells overexpressing $\alpha_v\beta_3$ integrin[21,22]. The amount of SCNPs internalised by cells decreased effectively by pre-treatmenting with free cRGDfK (20 μM), which strongly supported a mechanism, wherein the SCNPs were internalised via receptor-mediated endocytosis (Fig. 5f). PTX is a non-fluorescent anticancer drug, so a fluorescent dye cyanine 5.5 (Cy5.5) NHS ester was chosen to label SCNPs@PTX by reacting with the free amine groups on $\beta$-CD-NH$_2$. As shown in confocal laser scanning microscopy (CLSM) image (Fig. 5b), red fluorescence arising from Cy5.5 was observed in the cytoplasm after incubation HeLa cells with Cy5.5-labelled SCNPs@PTX for 8 h. However, the fluorescence intensity decreased significantly by pre-treating the cells with free cRGDfK (Supplementary Fig. 30), further confirming the targeting ability of SCNPs@PTX.

PTX is a cytoskeletal drug that targets tubulin by binding to the N-terminal 31 amino acids of the tubulin subunit, thereby stabilising the microtubule polymers and inhibiting disassembly (Fig. 5a)[23,24]. The intracellular GSH-responsive drug release was verified by detecting the mechanical properties of the cells after different treatments including reduced Young's modulus (Fig. 5d and Supplementary Fig. 31) and indentation depth (Fig. 5e and Supplementary Fig. 32) using atomic force microscopy measurements. Compared with the control group, the distribution of reduced Young's modulus for the cells treated with SCNPs@PTX (PTX, 50 nM) became much broader, and the reduced Young's modulus increased from $0.98 \pm 0.08$ kPa to $2.80 \pm 0.11$ kPa, which was much higher than that of SCNPs@PTX ($1.16 \pm 0.08$ kPa), firmly validating their chemotherapeutic efficacy of SCNPs@PTX. In terms of indentation depth, opposite trends were observed, greater cell stiffness resulted in low indentation depth. These two complementary parameters demonstrated that SCNPs are promising DDSs to deliver hydrophobic PTX.

The synergistic therapeutic effect of SCNPs@PTX with photothermal ablation from polyrotaxane and chemotherapy from PTX against HeLa cells was further evaluated. The half growth inhibition concentration (IC$_{50}$) was determined to be $5370 \pm 638$, $76.1 \pm 8.82$, and $251 \pm 38.3$ nM, for SCNPsCC@PTX, free PTX, and SCNPs@PTX, respectively (Fig. 5g). Due to the lack of GSH-responsiveness, the cytotoxicity of SCNPsCC@PTX was about 70-fold less than that of PTX. The chemotherapeutic efficacy of SCNPs@PTX was also lower than free PTX at equivalent PTX concentrations, because time-dependent drug release characteristics of SCNPs@PTX led to a delay in therapeutic efficacy, thus mitigating cytotoxicity in vitro. Upon irradiation (671 nm, 0.1 W cm$^{-2}$) for 3 min, the IC$_{50}$ value of SCNPs@PTX decreased to $164 \pm 20.7$ nM, mainly attributed to the accelerated drug release from SCNPs@PTX triggered by the photothermal effect, because the photothermal effect was insufficient to kill the cells mentioned above (Supplementary Fig. 33). Notably, the IC$_{50}$ value of SCNPs@PTX was remarkably decreased to $33.4 \pm 5.76$ nM when the laser power increased to 0.5 W cm$^{-2}$, showing excellent synergistic effect between chemotherapy and PTT. It should be noted that the IC$_{50}$ value increased to $635 \pm 77.2$ nM for the cells pre-treated with cRGDfK. This observation could be rationalised by the blockage of $\alpha_v\beta_3$ integrin receptors by free cRGDfK, which resulted in reduced cellular uptake of SCNPs@PTX, further demonstrating the targeting ability of SCNPs@PTX.

After treatment with different formulations, calcein acetoxymethyl ester and propidium iodide (PI) were utilized to stain the cells to visualise viable and dead cells, respectively (Supplementary Fig. 34). Compared with the negative control group, HeLa cells treated with either the NIR laser, PTX, SCNPs + laser, or SCNPs@PTX exhibited moderate cell death, indicating that such treatments did not compromise cell viability significantly. Obvious cell death was achieved by treating HeLa cells with SCNPs@PTX followed by NIR laser irradiation (0.5 W cm$^{-2}$). The cell viability was further examined by flow cytometry to elucidate the relative contributions of PTT and chemotherapy towards the cytotoxicity of the combination therapy (Fig. 5h). HeLa cells were treated with PTX, SCNPsCC@PTX, SCNPs@PTX, SCNPs + laser (0.5 W cm$^{-2}$), and SCNPs@PTX + laser (0.5 W cm$^{-2}$), and then labelled with PI and Annexin V. As shown in Fig. 3h, thermo-chemotherapy induced significant late apoptosis/necrosis (97.7%) as compared with the group treated with SCNPs + laser (51.6%) or PTX (39.6%), emphasising the cytotoxic effect of SCNPs@PTX against HeLa cells was boosted by PTT.

**In vivo PA and PET imaging.** The size of SCNPs located in the optimal range for tumour penetration via the so-called EPR effect, in which NPs are prone to be taken up arising from the leaky vasculature of tumour[25–27]. Moreover, the PEG segments on the shell of SCNPs form "brush-like" superstructures, preventing proteins from penetrating the surface and circumventing secondary adsorption onto the outer surface of the PEG layer, which is favourable to prolong their circulation time in blood, thereby promoting the accumulation of SCNPs in tumour tissue[28,29]. The pharmacokinetics of SCNPs@PTX and free PTX were evaluated in normal mice at various time points post intravenous (i.v.) injection (Supplementary Fig. 35). The circulation half-life and the area under the curve of SCNPs@PTX were much higher than those of free PTX, confirming that SCNPs@PTX exhibited longer blood retention. Biodistribution analysis indicated that SCNPs@PTX exhibited a 4.41-fold increase in tumour accumulation as compared to free PTX (Supplementary Fig. 36), owing to both EPR effect and active integrin-receptor-targeting effect.

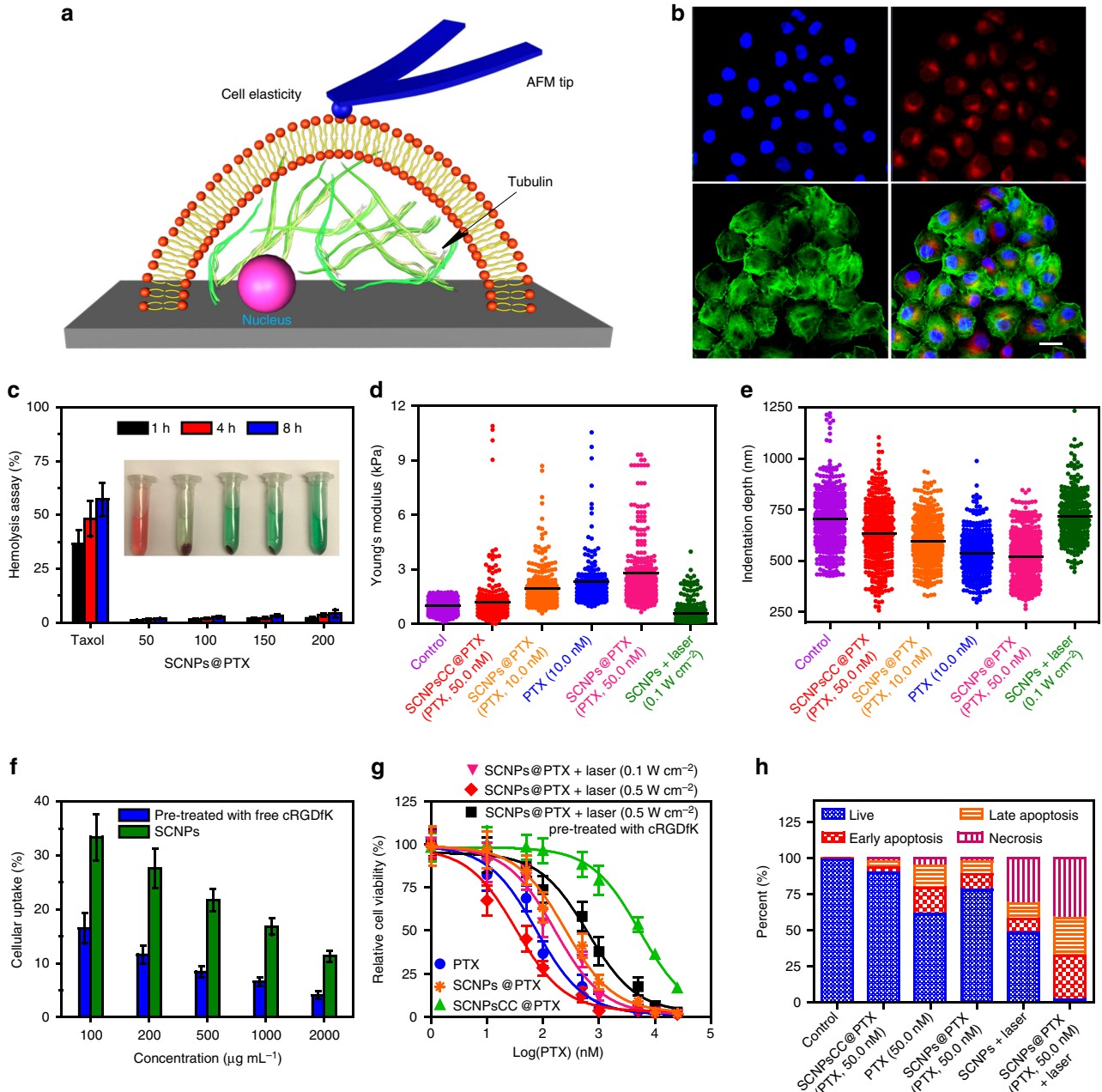

**Fig. 5** In vitro thermo-chemotherapy. **a** Schematic of AFM method used in the measurements of cell mechanical properties. **b** CLSM images of the HeLa cells cultured with Cy5.5-labelled SCNPs@PTX. Blue fluorescence shows nuclear staining with Hoechst 33342; red fluorescence shows the location of Cy5.5-labelled SCNPs@PTX; green fluorescence shows β-actin staining with FITC-phalloidin. Scale bar is 20 μm. **c** Haemolysis rates of PTX (50 μg mL$^{-1}$) and SCNPs@PTX at various concentrations (50, 100, 150, and 200 μg PTX mL$^{-1}$). Inset: the corresponding image of RBCs treated with PTX and SCNPs@PTX at various concentrations for 8 h. Histogram of **d** reduced Young's modulus and **e** indentation depth for all data collected from six different groups. The irradiation time was 3 min. **f** Cellular uptake of SCNPs@PTX under various concentrations in the absence and presence of free cRGDfK (20 μM). **g** In vitro cytotoxicity of different formulations towards HeLa cells. **h** Flow-cytometric analysis of Annexin-V/PI staining of HeLa cells after different treatments. The laser density was 0.5 W cm$^{-2}$, and the irradiation time was 3 min. Data are expressed as means ± s.e.m. ($n = 5$)

As a newly emerging imaging technique, PA imaging offers high spatial resolution and allows for deep tissue penetration[30,31]. SCNPs with high NIR-absorption and photothermal effect can be explored as an efficient contrast agent for PA imaging (Supplementary Figs. 24, 25). To demonstrate the imaging capability of SCNPs, we evaluated HeLa tumour-bearing mice by PA imaging. The PA intensity gradually increased over time in the tumour areas post-injection of SCNPs. For example, the PA intensity in the tumour was ~1.68-fold higher than the background signal at 2 h post-injection (Fig. 6a), indicating that SCNPs rapidly accumulated in the tumour tissue. The maximum PA signal in the tumour was found at 24 h post-injection, which was 2.76-fold higher than that of the control (Fig. 6d). Notably, the signal was maintained even at 48 h post-injection, and the tumour tissue around the blood vessels was bright. Additionally, 3D PA images at such a time points also evidently indicated the PA signals arose from the regions both inside and outside of the blood vessels (Fig. 6b). In vivo PA spectrum in the tumour site

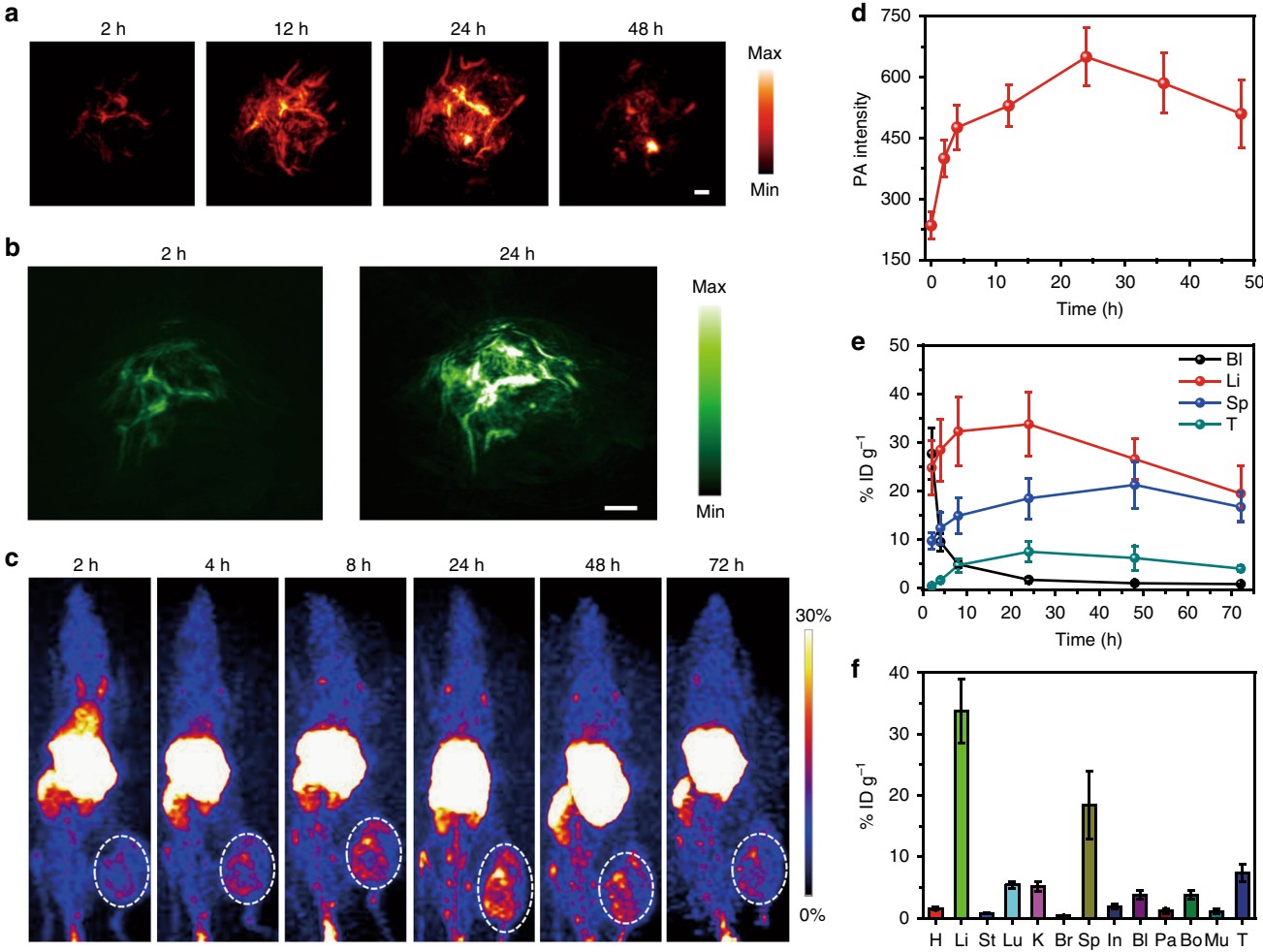

**Fig. 6** In vivo PA and PET imaging. **a** Representative PA maximum imaging projection (MIP) and **b** 3D images of tumour in a living mouse after systemic administration of SCNPs through i.v. injection. Scale bar is 2 mm. **c** Decay-corrected whole-body coronal PET images of HeLa tumour-bearing mice at 2, 4, 8, 24, 48, and 72 h after i.v. injection of 150 μCi of [64]Cu SCNPs. **d** Quantification of PA intensities at 671 nm as a function of post-injection time of SCNPs ($n = 3$). **e** Time-activity curves quantified based on PET images ($n = 3$). **f** Biodistribution of the [64]Cu SCNPs in mice bearing HeLa tumours at 24 h post-injection ($n = 3$). Data are expressed as means ± s.e.m

displayed high absorbance around 670 nm, which further confirmed the improvement of PA signal in the tumour was attributed to the accumulation of SCNPs (Supplementary Fig. 37).

To provide more precise and detailed anatomical or biological information for clear diagnosis and imaging, PET imaging was employed to quantitatively analyse the dynamic biodistributions and accumulations of SCNPs in the main organs during treatment[32,33]. The radialabeling was effectively maintained with <1% detachment of radionuclide from SCNPs, even after culturing the [64]Cu SCNPs@DOTA in PBS containing 10% mouse serum for 48 h (Supplementary Fig. 38), confirming the high labelling stability of the [64]Cu SCNPs@DOTA. HeLa tumour-bearing mice were imaged at various time points post i.v. injection of [64]Cu SCNPs@DOTA (150 μCi). The percentage injected dose per gram (% ID $g^{-1}$) of the [64]Cu SCNPs@DOTA in heart decreased gradually due to blood clearance (Fig. 6c). Interestingly, [64]Cu SCNPs@DOTA showed an efficient time-dependent tumour accumulation after injection (Fig. 6e). Quantitative region-of-interest analysis of these images revealed 1.5% ID $g^{-1}$ uptake of [64]Cu SCNPs@DOTA in the tumour at 4 h post-injection, which increased to 7.4% ID $g^{-1}$ at 12 h and maintained at 6.1% ID $g^{-1}$ at 48 h post-injection. For the liver, an increase in radioactivity signal was detected from 24.7 ± 5.62 % ID $g^{-1}$ at 2 h post-injection to 32.3 ± 7.16 % ID $g^{-1}$ at 8 h post-injection,

because the [64]Cu SCNPs@DOTA were captured by the reticuloendothelial system. Ex vivo biodistribution studies at 24 h post-injection indicated that the uptakes of [64]Cu SCNPs@DOTA by heart, liver, stomach, lung, kidneys, brain, spleen, intestine, bladder, pancreas, bone, and muscle were 1.61 ± 0.29, 33.7 ± 5.22, 0.83 ± 0.11, 5.46 ± 0.63, 5.17 ± 0.78, 0.42 ± 0.10, 18.4 ± 5.50, 1.90 ± 0.45, 3.86 ± 0.73, 1.18 ± 0.39, 3.81 ± 0.57, and 1.17 ± 0.38 % ID $g^{-1}$, respectively (Fig. 6f).

**In vivo thermo-chemotherapy.** Prior to the evaluation of in vivo anti-tumour activity, it is crucial to determine that SCNPs@PTX is safe for i.v. administration. As shown in Fig. 5c, SCNPs@PTX did not give rise to apparent haemolysis in the test concentration range, and the haemolysis percentage rose only slightly (1.1–4.3%) as concentration increased to the range of 50−200 μg PTX $mL^{-1}$. For PTX, the haemolysis percentage reached 57.1% at a concentration of 50 μg PTX $mL^{-1}$ after 8 h incubation, indicating that Cremophor EL used for PTX solubilisation is highly toxic for erythrocytes. In the present work, the zeta potential of SCNPs was 6.75 mV, thus unlikely to interact with negatively charged membranes of red blood cells. The results indicated that SCNPs@PTX based on amphiphilic copolymer and host–guest complex were favourable for in vivo application.

The MTD values of free PTX, blank SCNPs, and SCNPs@PTX were examined after a single i.v. injection into healthy mice. MTD value was defined as the highest dose leading to <15% body weight loss with no treatment-related death. Compared with untreated mice, no obvious weight change and no deterioration in health was observed for the mice treated with SCNPs, even at an extremely high dose of 1000 mg kg$^{-1}$, indicating excellent biocompatibility of our supramolecular DDS. For the mice treated with PTX, only a dose of 40.0 mg kg$^{-1}$ of PTX was tolerated, while one and three mice died in the groups receiving 50.0 and 60.0 mg PTX kg$^{-1}$, respectively (Table 2). Some mice in groups treated with PTX at 30.0 mg kg$^{-1}$ suffered diarrhoea and displayed lethargy. For the SCNPs@PTX-treated group, the MTD value increased to 80.0 mg PTX per kilogram without death. During the observation period, maximum weight loss appeared at 6 days after administration, while the majority of the groups regained the lost weight. As confirmed by biodistribution studies, the increase in tumour accumulation and the decrease in exposure of SCNPs@PTX to spleen, kidneys, and heart was one possible mechanism that may explain why the MTD value of SCNPs@PTX was much higher than that of free PTX (Supplementary Fig. 36). Moreover, the shell crosslinking strategy dramatically slowed down the release of PTX from SCNPs@PTX, thus decreasing its toxicity towards the main organs, such as the liver and lung, which was another important explanation for the supramolecular nanomedicine mediated increase of MTD.

Inspired by the exciting results of our in vitro thermo-chemotherapy study, the in vivo anti-tumour therapeutic efficacy was investigated by using HeLa tumour-bearing mice. The mice were randomly divided into six groups: (I) PBS, (II) SCNPs@PTX (PTX, 20 mg kg$^{-1}$), (III) Abraxane (PTX, 20 mg kg$^{-1}$), (IV) SCNPs@PTX (PTX, 60 mg kg$^{-1}$), (V) SCNPs + laser, and (VI) SCNPs@PTX (PTX, 60 mg kg$^{-1}$) + laser. Compared with their MTD values, the injected doses here were reduced in order to reduce their potential side effect and enhance the quality of life for the mice. In vivo photothermal effects of SCNPs@PTX on the mice bearing HeLa tumours were evaluated at 24 h post-injection, showing a rapid temperature increase from 29.8 to 51.3 °C upon laser irradiation, which was enough for heat-induced tumour inhibition (Supplementary Figs. 39, 40). As shown in Fig. 7a, the average tumour volume in the PBS group increased rapidly, suggesting the tumour growth was not affected. Treatment with SCNPs@PTX at a low dosage of 20 mg PTX kg$^{-1}$ only delayed the tumour growth, and recurrence occurred quickly. As a chemotherapeutic agent that has already proved to be effective against many tumours, the therapeutic performance of Abraxane was not satisfactory after administration of a single-dose injection (PTX, 20 mg kg$^{-1}$). Compared with the Abraxane-treated group, somewhat better anti-tumour efficacy was observed for the mice receiving a high dose of SCNPs@PTX (PTX, 60 mg kg$^{-1}$). For PTT, the tumour volume showed an obvious decrease immediately, whereas recurrent tumour growth appeared in the later period at 24 days. In contrast with the other groups, the formulation of SCNPs@PTX (PTX, 60 mg kg$^{-1}$) followed by laser irradiation showed excellent synergic effects, far exceeding the efficacy from PTT or chemotherapy alone. SCNPs@PTX-mediated chemotherapy and laser-irradiation-active PTT completely ablated the tumours without any recurrence during the experimental period after a single-dose injection. The reason for this was that a higher temperature activated the "switch" of the nanocapsules to release the preloaded drugs, which diffused into a greater effective area, resulting in continuous chemotherapeutic effects, thus inhibiting tumour recurrence and avoiding the necessity for multiple injections. Notably, Kaplan–Meier survival curves indicated that the group administered with SCNPs@PTX (PTX, 60 mg kg$^{-1}$) + laser had a 100% survival rate during the

### Table 2 Treatment response for MTD study

| | Dose (mg kg$^{-1}$)[a] | No. of nude mice | Max % wt loss (day)[b] | Death (day) |
|---|---|---|---|---|
| PTX | 20.0 | 6 | 3.43 (3) | 0 |
| | 30.0 | 6 | 7.35 (7) | 0 |
| | 40.0 | 6 | 10.4 (6) | 0 |
| | 50.0 | 6 | 16.2 (7) | Culled[c] |
| | 60.0 | 6 | 18.7 (6) | 1 (3) |
| | | | | 1 (5) |
| | | | | Culled |
| SCNPs | 100 | 6 | Increase[d] | 0 |
| | 200 | 6 | Increase | 0 |
| | 400 | 6 | Increase | 0 |
| | 800 | 6 | 1.17 (3) | 0 |
| | 1000 | 6 | 1.04 (5) | 0 |
| SCNPs@PTX | 20.0 | 6 | 0.87 (4) | 0 |
| | 40.0 | 6 | 4.16 (5) | 0 |
| | 60.0 | 6 | 7.74 (7) | 0 |
| | 80.0 | 6 | 10.6 (6) | 0 |
| | 100 | 6 | 16.6 (7) | 1 (2) |
| | | | | Culled |

[a]The groups (PTX and SCNPs@PTX) containing PTX are based on mg PTX per kg, the blank control (SCNPs) is based on mg SCNPs per kg
[b]Maximum percent body weight loss
[c]Animals culled due to exceeding 15% body weight loss
[d]No decrease in body weight was observed during the experimental period

experimental period, whereas the median survival was determined to be 30, 37, 51, 58, and 75 days for the mice in the groups I, II, III, IV, and V, respectively (Fig. 7b). The H&E staining of the tumour tissue from the mice treated with SCNPs@PTX (PTX, 60 mg kg$^{-1}$) + laser displayed more-severe damage than those from pure chemotherapy or PTT groups. The tumour cells suffered the most serious fibrosis compared with the other groups, in which abundant karyolysis and a high level of necrosis were observed (Supplementary Fig. 41). It should be noted that the combination of PTT and chemotherapy also completely ablated the HeLa tumours with larger size (~220 mm$^3$) without recurrence (Supplementary Figs. 45, 46), firmly verifying superior anti-tumour efficacy of our supramolecular nanomedicine.

The potential toxicity of this supramolecular theranostic nanomedicine in vivo is a crucial issue for practical applications. It is expected that, if some SCNPs@PTX is captured by normal tissues, only a small fraction of the encapsulated drug is released in the absence of a NIR laser, thus successfully avoiding unwanted side effects. Histology analysis of the major organs (heart, liver, spleen, lung, and kidneys) stained with H&E were conducted 10 days after treatment to evaluate the potential toxicity towards the major organs. No obvious tissue damage or inflammatory lesion was observed in all major organs (Supplementary Fig. 42), verifying no/very-low systemic toxicity of SCNPs@PTX at our tested dose. Moreover, negligible changes in body weight (Supplementary Fig. 43) and haematological tests (Supplementary Fig. 44) were observed in the group treated with SCNPs@PTX (PTX, 60 mg kg$^{-1}$) + laser (Supplementary Fig. 43), suggesting that no noticeable systemic toxicity of the thermo-chemotherapy.

Encouraged by the superior anti-tumour results on xenograft tumour model, we anticipated that our supramolecular nanomedicine is an ideal candidate to treat highly aggressive 4T1 breast cancer by taking advantage of chemotherapy and PTT. The synergistic thermo-chemotherapy was confirmed by MTT assay (Supplementary Fig. 47), in which the IC$_{50}$ value was determined to be 23.1 ± 5.87 nM for the 4T1 cells treated with SCNPs@PTX followed by laser irradiation (671 nm, 0.5 W cm$^{-2}$) for 3 min,

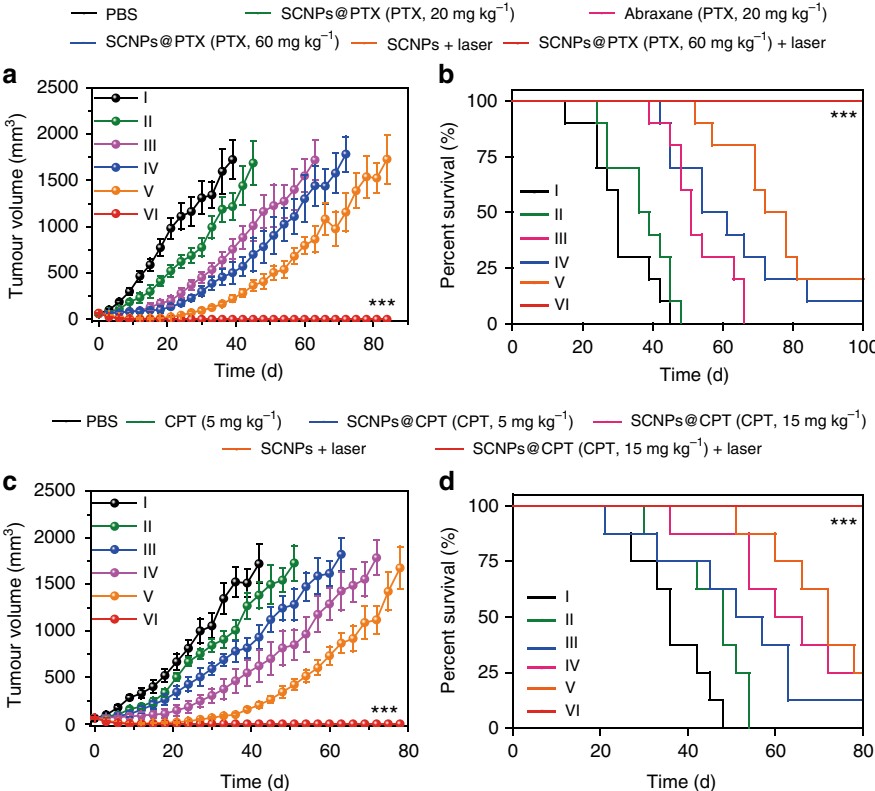

**Fig. 7** In vivo thermo-chemotherapy on xenograft tumours. **a** Tumour volume changes and **b** Kaplan–Meier survival curves of the mice bearing HeLa xenografts treated with different formulations after one injection ($n = 10$). **c** Tumour volume changes and **d** Kaplan–Meier survival curves of mice bearing A549 xenografts treated with different formulations after one injection ($n = 8$). The laser density was 0.5 W cm$^{-2}$, and the irradiation time was 5 min. Data are expressed as means ± s.e.m., ***$P < 0.001$

which was much lower than those of PTX (IC$_{50}$ = 87.9 ± 10.1 nM), SCNPs@PTX (IC$_{50}$ = 211 ± 32.5 nM), and SCNPs@PTX + laser (0.1 W cm$^{-2}$, 3 min) (IC$_{50}$ = 141 ± 22.3 nM). 4T1 tumours were orthotopically inoculated in the mammary fat pads of the mice to produce spontaneous lung metastatic breast cancer, in order to validate the merit of SCNPs@PTX on the antimetastasis efficacy.

The mice were randomly separated into six groups ($n = 8$) and administered with (I) PBS, (II) SCNPs@PTX (PTX, 20 mg kg$^{-1}$), (III) Abraxane (PTX, 20 mg kg$^{-1}$), (IV) SCNPs@PTX (PTX, 60 mg kg$^{-1}$), (V) SCNPs + laser, and (VI) SCNPs@PTX (PTX, 60 mg kg$^{-1}$) + laser, respectively. Compared with the mice treated with PBS, very-weak anti-tumour efficacy was achieved using SCNPs@PTX (PTX, 20 mg kg$^{-1}$) with low dosage (Fig. 8a). In the meantime, only moderate tumour inhibitions were found for the groups administered with Abraxane (PTX, 20 mg kg$^{-1}$), SCNPs@PTX (PTX, 60 mg kg$^{-1}$), and SCNPs + laser, about 26.4%, 39.9%, and 50.6% tumour reduction was achieved compared with the PBS group. The strongest anti-tumour effect was observed for the group treated with SCNPs@PTX (PTX, 60 mg kg$^{-1}$) followed by laser irradiation, as evidenced by the nearly complete ablation of the primary orthotopic tumours, attributing to the synergistic efficacy of PTT and chemotherapy (Fig. 8e). In sharp comparison with the other five groups, the median survival was remarkably increased for the mice in group VI (Fig. 8b), further confirming the excellent anti-tumour performance. The weight of each tumour mass was measured at the end of the treatment (Supplementary Fig. 48), which also indicated that the combination of PTT and chemotherapy was the most effective in suppressing tumour growth. H&E staining and immunohisto-chemical staining (Ki-67) of the tumour sections confirmed significantly enhanced necrosis in the thermo-chemotherapy group compared with the control groups (Fig. 8f). Importantly, negligible changes in body weight were observed during the treatment (Supplementary Fig. 49), suggesting low systemic toxicity of the thermo-chemotherapy resulting from the smart design.

Furthermore, the antimetastatic activity of these formulations was assessed by analysing the number of surface lung metastases (Supplementary Figs. 50, 55), tumour coverage percentage (Fig. 8d), and subsequent analysis of lung H&E tissue sections (Fig. 8h). As shown in Fig. 8c, g, the average metastatic nodules on the lung surface were 8.13, 8.00, 5.36, 5.25, and 6.75 for the mice in groups I, II, III, IV, and V, respectively, indicating that only slight or moderate suppression of lung metastasis was achieved by chemotherapy or PTT alone. Excitingly, the thermo-chemotherapy effectively reduced metastatic nodules: only one or two tumour nodules were detected on the lungs. This observation was further validated by analysing the proportion of the metastasis area to the whole lung, with only 0.620% of lungs occupied by tumours after thermo-chemotherapy (Fig. 8d), which was much lower than those of the other treatments (12.9, 12.6, 6.48, 4.77, and 6.93% for the mice in groups I, II, III, IV, and V, respectively). The metastatic lung tumours were also directly visualised by PET/CT imaging using $^{18}$F-fluorodeoxyglucose (FDG) as a radiotracer. PET imaging clearly showed extensive tumour burden in lung for the randomly picked mice administered with PBS (Fig. 8i and Supplementary Figs. 56–61). Although the metastatic nodules decreased after chemotherapy (II, III, IV) or PTT (V) alone, high FDG signals in lung were still found, suggesting the weak antimetastasis capability of these administrations. Ascribing to its excellent anti-tumour efficacy,

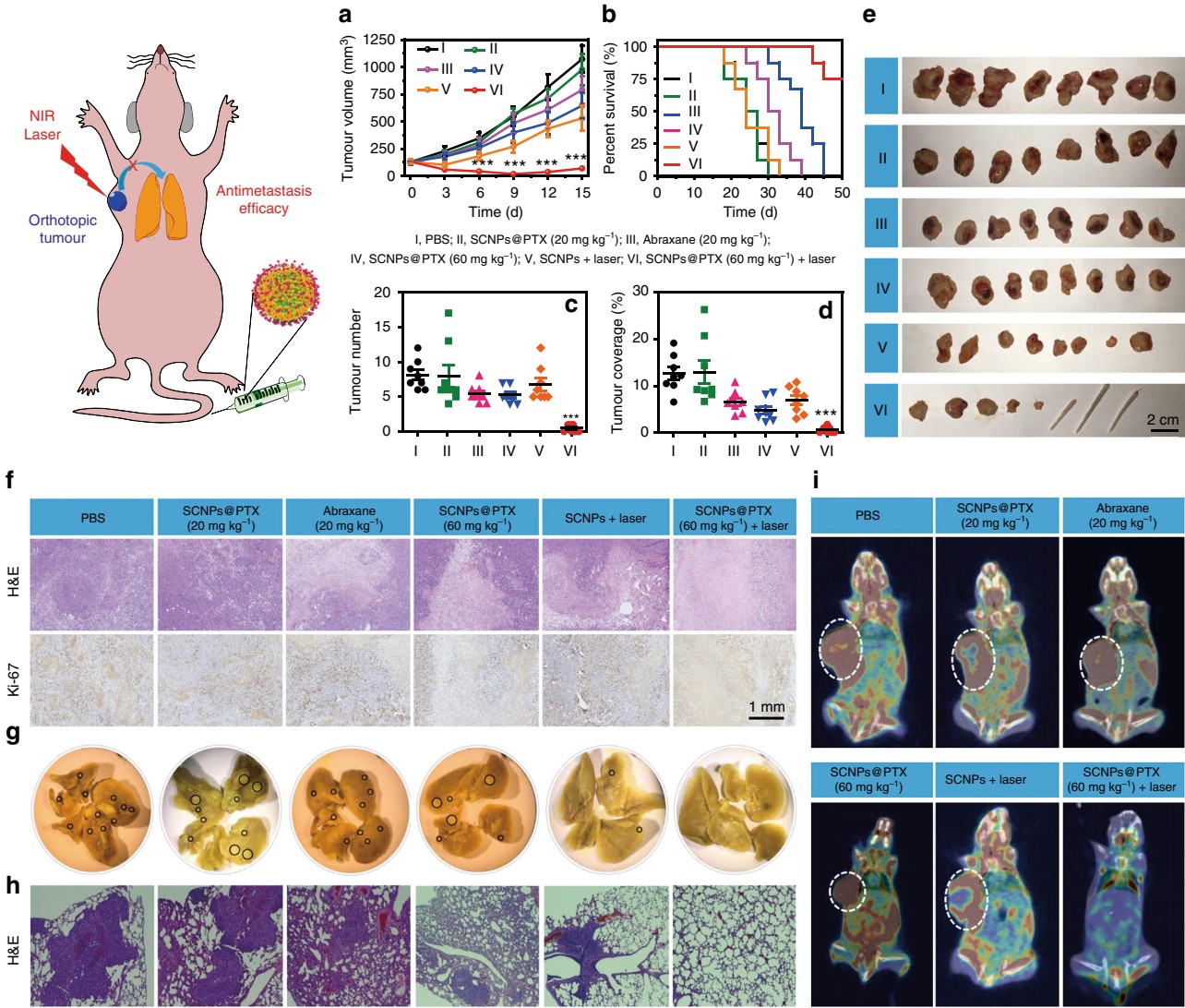

**Fig. 8** Treatment of orthotopic breast cancer and inhibitory effects on lung metastasis. **a** Tumour volume changes and **b** Kaplan–Meier survival curves of the mice bearing orthotopic 4T1 breast tumours treated with different formulations after one injection ($n = 8$). **c** The numbers of tumour nodules present on the lung surface from each group. **d** Tumour coverage percentage in the lungs from each group. **e** Photo images of the orthotopic tumours harvested from the mice treated with different formulations. **f** H&E and Ki-67 staining of the tumour tissues from each group. **g** Representative images of the lungs excised from each group. The black circles denote the visually detected metastatic nodules in each lung tissue. **h** Histological examination of metastatic lesions in lung tissues from each group after H&E staining. **i** PET/CT images of the mice treated with different formulations at the 14th day post-injection. Arrows indicates possible metastatic tumours. The laser density was 0.5 W cm$^{-2}$, and the irradiation time was 5 min. Data are expressed as means ± s.e. m., ***$P < 0.001$

the metastatic nodules were barely observed in the mice receiving thermo-chemotherapy. Thus, the combination of PTT and chemotherapy not only suppressed orthotopic primary tumour growth, but also successfully inhibited lung metastasis more effectively than either chemotherapy or PTT alone due to the unique properties of the supramolecular nanomedicine. Finally, to demonstrate the robust features of our supramolecular platform, we utilized SCNPs to encapsulate CPT with a loading content of 43.2% (Supplementary Fig. 62 and Supplementary Table 2). In vitro and in vivo investigations demonstrated that SCNPs@CPT possessed excellent anti-tumour performances against A549 xenograft model and orthotopic 4T1 breast cancer, greatly inhibiting tumour recurrance and metastasis (Fig. 7c, d, Supplementary Tables 2–3 and Supplementary Figs. 62–90). Combined, these results proved that SCNPs@PTX and SCNPs@CPT might serve as safe and efficient supramolecular nanomedicines for in vivo thermo-chemotherapy, which was

attributed to the smart polyrotaxane delivery system with sophisticated topological structure. This supramolecular theranostic platform provides a blueprint to guide the design of the next generation of nanomedicines for safe and effective cancer treatment.

## Methods

**Preparation of drug-loaded SCNPs**. For the preparation of drug-loaded SCNPs, one-pot synthesis method was utilized. PDI-PCL-*b*-PEG-RGD⊃*β*-CD-NH$_2$ (20.0 mg) was dissolved in 10 mL DMF in a 250 mL round bottom flask and allowed to stir for 1 h at room temperature. A stock solution of PTX or CPT was added into the solution and allowed to stir for additional 1 h. To this solution, deionized water (50 mL) was added dropwise via a syringe pump over a period of 6 h. In situ crosslinking was performed by adding a stock solution of NHS-SS-NHS. The mixture was allowed to stir for an additional 24 h at room temperature to afford drug-loaded SCNPs. The solution was dialysed against deionized water for 1 d in a presoaked dialysis tubing (MWCO 3 kDa) to remove the residual DMF, NHS byproducts, and free drug. NHS-SS-NHS first diffused into the NPs driven by electrostatic interactions between anionic NHS-SS-NHS and cationic amine groups

followed by -NHS/NH₂- coupling reaction, effectively inhibiting inter-nanoparticle crosslinking.

**Dye labelling or DOTA labelling of the SCNPs**. Cy5.5-NHS (50.0 µg) was dissolved in 100 µL dimethylsulfoxide (DMSO) and the solution was added into the aqueous solution containing SCNPs@PTX (2 mL) in the presence of one drop triethylamine. After stirring at room temperature overnight, the byproducts and DMSO were eliminated by dialysis (MWCO 3 kDa) against deionized water to afford Cy5.5-labelled SCNPs@PTX. For the preparation of DOTA labelled SCNPs, DOTA-NHS (100 µg) was added directly into the aqueous solution containing SCNPs (2 mL) in the presence of one drop triethylamine. After stirring at room temperature overnight, the byproducts and DMSO were eliminated by dialysis (MWCO 3 kDa) against deionized water to afford SCNPs@DOTA. The samples were kept at 4 °C for further use.

**Cell Cultures**. HeLa, A549, and 4T1 cell lines were purchased from American Type Culture Collection (ATCC, Rockville MD). HeLa, A549, and 4T1 cells were cultured in Eagle's MEM, Dulbecco's Modified Eagle's Medium, and RPMI-1640 medium, respectively, containing 4 mM L-glutamine, 4500 mg L$^{-1}$ glucose, 10% foetal bovine serum (FBS), 100 units per ml penicillin, and 100 units per ml streptomycin, supplied by GE Life Sciences Co. Ltd. These cell lines have passed the conventional tests of cell line quality control methods (e.g., morphology, iso-enzymes, and mycoplasma). The cells grew as a monolayer and were detached upon confluence using trypsin (0.5% w/v in phosphate-buffered saline). The cells were harvested from cell culture medium by incubating in the trypsin solution for 5 min. The cells were centrifuged, and the supernatant was discarded. A 3.00 mL portion of serum-supplemented medium was added to neutralise any residual trypsin. The cells were resuspended in serum-supplemented medium at a concentration of $1.00 \times 10^4$ cells mL$^{-1}$. The cells were cultured at 37 °C and 5% CO₂.

**Evaluation of cytotoxicity**. The cytotoxicities of free drug (CPT or PTX), SCNPsCC@PTX (or SCNPsCC@CPT), and SCNPs@PTX (or SCNPs@CPT) against HeLa and A549 cells with/without laser irradiation were determined by MTT assays in a 96-well plate. The cells pre-treated with free cRGDfK (20 µM) for 30 min were used as a control. All solutions and mediums were sterilised by filtration (0.22 µm) before use. HeLa and A549 cells were seeded at a density of $1 \times 10^4$ cells per well in a 96-well cell culture plate, and incubated for 24 h for attachment. Then the cells were cultured with free drug (CPT or PTX), SCNPsCC@PTX (or SCNPsCC@CPT), and SCNPs@PTX (or SCNPs@CPT) at various concentrations for 24 h. For the PTT, the cells were incubated with SCNPs, SCNPs@PTX (or SCNPs@CPT) at different concentrations for 12 h, followed by laser irradiation for 3 mins, then the cells were further cultured for another 12 h. After washing the cells with PBS buffer, 20 µL MTT solution (5 mg mL$^{-1}$) was added to each well. The MTT solution was removed after 4 h of incubation, and the cells were washed with PBS for three times. A volume of 100 µL DMSO was added to each well to solubilise formazan crystals, and the absorbance of the formazan product was measured at 570 nm using a spectrophotometer (Bio-Rad Model 680). All experiments were carried out with five replicates.

**Evaluation of haemolysis**. Blood was collected from the nude BALB/c mice. The blood sample was diluted 10 times with PBS and centrifuged at 1500 rpm for 10 min. The centrifuged sample was washed with sterile PBS for five times by centrifugation and suction to isolate erythrocytes. The concentration of the resultant blood cells was adjusted to 2% (v/v). A 100 µL sample solution was added to the blood cells (1000 µL) and the mixture was incubated for 1 h, 4 h, and 8 h at 37 °C. The corresponding sample was centrifuged for 15 min at 2000 rpm. In order to assess the haemolytic activity, the released haemoglobin in supernatant was determined by measuring the absorbance at 541 nm. The percentage of haemolysis was determined as $(A_{sample} - A_0)/(A_{100} - A_0) \times 100\%$, where $A_{sample}$, $A_{100}$, and $A_0$ is the absorbance of the sample, the completely lysed red blood cells in distilled water, and zero haemolysis in PBS, respectively. All haemolysis assays were conducted with five replicates.

**Tumour model**. Female BALB/c nude mice (4 weeks old, ~20 g body weight) were purchased from Zhejiang Academy of Medical Sciences and maintained in a pathogen-free environment under controlled temperature. Animal care and handing procedures were in accordance with the guidelines approved by the ethics committee of Zhejiang University. All study protocols involving animals were approved by the Zhejiang University Animal Care and Use Committee. The female nude mice were injected subcutaneously in the right flank region with 200 µL of cell suspension containing $5 \times 10^6$ HeLa cells (or A549 cells). The tumours were allowed to grow to ~60 mm³ before experimentation. The tumour volume was calculated as (tumour length) × (tumour width)²/2.

**Investigations of pharmacokinetics and tissue distributions**. Mice received free PTX, free CPT, SCNPs@PTX, or SCNPs@CPT at a dose of 5 mg PTX (or CPT) per kg body weight by tail vein injection. Blood was collected by cardiac puncture at 15 min, 0.5 h, 1 h, 2 h, 4 h, 8 h, 12 h, and 24 h post-injection and kept in heparinized tubes. The plasma was separated from the cell fraction by centrifugation for 10 min at 1500 g, and then 1 vol of plasma was mixed with 2 vol of acetonitrile containing 1.00 mM DTT and vortexed for 3 min followed by centrifugation for 5 min at 12,000 g. The PTX (or CPT) concentration in the supernatant was measured by HPLC. Animals were sacrificed at 24 h post-injection. The main organs (liver, lung, spleen, kidney, and tumour) were excised and washed quickly with PBS. To 100 mg of organ tissue was added 250 µL of PBS, pH 7.4, containing 1.00 mM DTT and the mixture was homogenised by a Bertin tissue grinder at speed of 6000 s$^{-1}$ for 2 min. The homogenised mixture was mixed with 500 µL of acetonitrile and vortexed for 3 min. The mixture was then centrifuged at 12,000 g and the supernatant was collected. The drug concentration in the supernatant was measured by HPLC.

**In vivo imaging**. PA imaging, SCNPs aqueous solution (3.00 mg mL$^{-1}$, 100 µL) was i.v. injected into the HeLa tumour-bearing mice. PA imaging of tumour was performed on an Endra Nexus 128 PA tomography system (Endra, Inc., Ann Arbor, MI). Photothermal imaging was detected by using a SC300 infrared camera when the tumours were irradiated with 671 nm laser (0.5 W cm$^{-2}$) for 5 min.

SCNPs were labelled with radioactive copper ($^{64}$Cu) by mixing $^{64}$CuCl₂ with SCNPs@DOTA at 37 °C for 1 h under constant stirring. As detected by thin-layer chromatography, the radiolabeling yield of SCNPs@DOTA was determined to be as high as 98%. For PET imaging, $^{64}$Cu SCNPs@DOTA (150 µCi, 100 µL) was intravenously injected into the HeLa tumour-bearing mice. The mice were anaesthetised with isoflurane (1.0 ~ 2.0%) in oxygen delivered at a flow rate of 1.0 L min$^{-1}$. All PET imagings were carried out on Inveon small-animal PET scanner (Siemens, Erlangen, Germany) at different times post-injection. For each in vivo PET scan, 3D volumes of interest were drawn over the tumour and muscle on decay-corrected whole-body coronal images, which were analysed by Inveon Research Workplace. Standards were prepared and the tumour uptake was determined as percent of injected dose per gram of tissue.

**In vivo cancer treatment on xenograft tumour models**. HeLa tumour-bearing mice were randomly divided into six groups ($n = 10$), which received different formulations: PBS, Abraxane (PTX, 20 mg kg$^{-1}$), SCNPs@PTX (PTX, 20 mg kg$^{-1}$), SCNPs@PTX (PTX, 60 mg kg$^{-1}$), SCNPs + laser irradiation (671 nm, 0.5 W cm$^{-2}$, 5 min), or SCNPs@PTX (PTX, 60 mg kg$^{-1}$) + laser irradiation (671 nm, 0.5 W cm$^{-2}$, 5 min). Tumour volumes and body weight of the mice were measured every 2 days after treatments by using a caliper and an electronic balance, respectively. For the treatment of HeLa tumour with larger size (around 220 mm³), the irradiation time was extended to 10 min (0.5 W cm$^{-2}$) in order to enhance the anti-tumour efficacy.

A549 tumour-bearing mice were randomly divided into six groups ($n = 8$), which received different formulations: PBS, CPT (5 mg kg$^{-1}$), SCNPs@CPT (CPT, 5 mg kg$^{-1}$), SCNPs@CPT (CPT, 15 mg kg$^{-1}$), SCNPs with 671 nm laser irradiation (0.5 W cm$^{-2}$, 5 min), or SCNPs@CPT (CPT, 15 mg kg$^{-1}$) with 671 nm laser irradiation (0.5 W cm$^{-2}$, 5 min). Tumour volumes and body weight of the mice were measured every 2 days after treatments by using a caliper and an electronic balance, respectively.

**Treatment of orthogonal breast cancer and inhibitory effects of lung metastasis**. To confirm the anticancer efficacy against orthotopic breast cancer and antimetastasis effect in vivo, 4T1 cells ($5 \times 10^5$) were xenografted into the breast fat-pad of the mice. Anti-tumour treatments were performed when the tumours were around 100 mm³. The mice were randomly divided into several groups ($n = 8$) and treated with PBS, Abraxane (PTX, 20 mg kg$^{-1}$), SCNPs@PTX (PTX, 20 mg kg$^{-1}$), SCNPs@PTX (PTX, 60 mg kg$^{-1}$), SCNPs + laser irradiation (671 nm, 0.5 W cm$^{-2}$, 5 min), SCNPs@PTX (PTX, 60 mg kg$^{-1}$) + laser irradiation (671 nm, 0.5 W cm$^{-2}$, 5 min), CPT (5 mg kg$^{-1}$), SCNPs@CPT (CPT, 5 mg kg$^{-1}$), SCNPs@CPT (CPT, 15 mg kg$^{-1}$), SCNPs + laser irradiation (671 nm, 0.5 W cm$^{-2}$, 5 min), or SCNPs@CPT (CPT, 15 mg kg$^{-1}$) + laser irradiation (671 nm, 0.5 W cm$^{-2}$, 5 min). At the end time point, the mice received different formulations were killed, and their lung tissues were carefully removed, and photographed. The number of metastatic nodules from each group were counted and recorded to evaluate the inhibition of lung metastasis. Moreover, the lung tissues from each group after treatment were assessed by histological examinations to detect the metastatic lesions.

**Other methods**. Other information about syntheses, characterisations, in vitro studies, and in vivo investigations are given in Supplementary Information.

**Data availability**. All data are available from the authors upon reasonable request.

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

## Acknowledgements

This work was supported by the Intramural Research Program of the National Institute of Biomedical Imaging and Bioengineering, National Institutes of Health, National Natural Science Foundation of China (21674091, 21434005, 91527301, and 51673171), National Basic Research Program (2013CB834502), Zhejiang Provincial Natural Science Foundation of China (Grant LR16E030001), and the Open Project of State Key Laboratory of Supramolecular Structure and Materials.

## Author contributions

G.Y., Z.M., Q.F., F.H., and X.C. conceived and designed the research. Y.L. and O.J. performed the PET imaging studies. S.W. and Z.Y. conducted the PA imaging studies. X.F., X.T., and A.J performed the AFM investigations. G.Y., Y.L., F.Z., C.G., and Z.M. performed the in vitro and in vivo experiments. J.Y., L.S., B.H., and G.Y. characterised the polyrotaxane and analysed the data. G.Y., Z.Y., Z.M., B.Y., F.H., and X.C. co-wrote the paper.

## Additional information

**Competing interests:** The authors declare no competing financial interests.

