## [Peer Review File · Nature Communications]

Reviewers' comments:

Reviewer #1 (Remarks to the Author):

Nanoparticles for combination of chemotherapy and photothermal therapy have demonstrated enhanced efficacy in various *in vivo* settings. In "Polyrotaxane Based Supramolecular Theranostics", Dr. Xiaoyuan Chen, Dr. Feihe Huang, and their colleagues, prepared nanoparticles from polyrotaxane polymers to deliver anticancer agents and photosensitizers to tumors, while following their accumulation by PET. The theranostic nanoparticles showed potent antitumor effects at high chemotherapy and light doses. However, the novelty of the strategy is not clear. Moreover, the characterization of the materials should be further evaluated to confirm the proposed structure. In addition, the antitumor effects should be evaluated in challenging tumor models. Therefore, I suggest the following revisions for improving the quality of the manuscript:

1. The extent of cross-linking of the NHS-SS-NHS should be studied. Indeed, the length of the NHS-SS-NHS appears to be short for intermacromolecular bridging, as the distance between the amines in cyclodextrins appears to be larger than the length of NHS-SS-NHS. Moreover, the possibility for binding with the cyclodextrin in the same polymer unit should be demonstrated. Finally, the residual NHS or COOH groups after the cross-linking should be quantified.
2. The stability of the particles in blood should be evaluated. From the PET results, the nanoparticles are showing extensive accumulation in liver and spleen, which may suggest that they are not functioning as designed. Particularly, I am worried about the aggregation and opsonization of the particles in blood.
3. What is the effect of the rotaxanes in the polymers on drug release and stability? Studying various rotaxane amounts in the polymers will clarify this point.
4. Besides the drug release, the disassembly of the particles at various GSH conditions should be evaluated. Particularly, I am interested to see if the photothermal induced cellular damage provokes an acceleration of the drug release and particle disruption.
5. The PET imaging results on the accumulation of the particles in tumors should be correlated with the amount of paclitaxel being delivered to the tumor to that the particles avoid the premature drug release *in vivo*.
6. While the authors used large tumors for PET imaging, the starting tumor size in the antitumor activity experiment is too small. Because a major issue for photothermal therapies is the poor light penetration into deep regions, the efficacy of the nanoparticles should also be studied in larger tumor masses to assess the potential of the system. Moreover, the authors are using subcutaneous HeLa xenografts for the evaluation of the antitumor effects. Such tumor model is not appropriate. Study the nanoparticles in orthotopic or metastatic tumors in mice with competent immune systems is more significant.
7. The novelty of the study should be clarified. Based on the authors' explanations, the main novelty of the study is avoiding premature drug release from the carriers. However, there are already several formulations that can avoid premature drug release and control the delivery of drugs. From the current manuscript, my impression is that the authors are just putting together various strategies, such as host-guest assembly with rotaxanes [Li, J., et al. *Adv. Drug Deliv. Rev.*, 60 (2008) 1000-1017; etc.], combination of chemotherapy and photothermal therapy [Zou, L, et al. *Theranostics* 6 (2016) 762–772; etc.], active targeting by using cRGD peptide [Park, J. H., et al. *J. Control. Release*, 95 (2004)

579-588; etc.], into a complicated design.

Minor point:

1. Instead of using I, II, III, IV,... in the figures, it would be much clear to use the names of the formulations.

Reviewer #2 (Remarks to the Author):

The paper "polyrotaxane based supramolecular theranostics" reports the preparation, characterization and in vitro/in vivo evaluation of a supramolecular self-assembly drug delivery system for the RGD-targeted delivery of anticancer drugs, like paclitaxel and camptothecin.

The work is relevant and contains many experimental data supporting the efficacy of the proposed nanosystems. The study fit with the scope of the journal but before publication the authors should address the following points mainly related to the characterization and identification of the supramolecular structure:

- The syntheses of PDI-PCL, PDI-PCL-b-PEG, PDI-PCL-b-RGD=beta-CD-NH₂ are briefly described in the "results and discussion" section but are missed in the "material and methods" section. So no details are given about these relevant procedures.
- Some figures are wrongly cited in the body text, i.e. pag 5 Fig. 1 should be scheme 1, pag. 6 Fig. 2a should be Fig.1a, pag. 9 Fig. 3c should be Fig. 2c, etc.
- The Figures are sometimes not incrementally cited in the text, there are many supplemental figures not cited in the text and at least it is necessary to mention what can be found in the supplementary file.
- There are not experimental conditions on how the samples have been analysed by GPC, solvent, column and especially the standards used for column calibration.
- GPC cannot be used to estimate the number of cyclodextrin included in PDI-PCL-b-RGD=beta-CD-NH₂ because the MW estimation is accurate only when the standards are similar to the analysed sample. Here the standards are not specified but I don't think a series of PDI-PCL-b-RGD=beta-CD-NH₂ with different cyclodextrin units have been used. GPC separation is based on hydrodynamic volume and this might change dramatically with the addition of even a couple of cyclodextrins. So the number of seven cyclodextrins is not experimental validated.
- Regarding the preparation of polyrotaxanes, it is very important to demonstrate that the cyclodextrin is really pierced by PCL segment instead of having a conformation in which PCL forms a hairpin in the cavity of cyclodextrin. Also the last conformation will be stable and give similar results in the experiments used to characterize the polyrotaxane. Eventually only organic solvents may destabilize such conformation. Furthermore it has to be considered that to reach the PCL segment the cyclodextrin has to be pierced firstly by hydrophilic and flexible PEG starting from the maleimide group, a sequence of events which seems not so favoured in comparison to direct interaction with PCL hairpins.
- It is know that maleimide reacts with amines and when thiol groups are not present such reaction proceed quite well. Since I suppose PEG-PCL-b-PEG-MAL have been incubated with an excess of

amino-cyclodextrin, I am wondering how such reaction has been prevented. The presence of amino groups should be verified by specific assay.

- NHS-SS-NHS can also reacts with hydroxyl groups of cyclodextrins and also this reaction will lead to a decrease of zeta potential because part of the NHS-SS-NHS molecules may bind an hydroxyl group at one side and the other NHS will be hydrolysed to COOH by water instead to react with another OH.

- NHS-SS-NHS is not anionic, please correct and adjust the discussion about electrostatic interaction of NHS-SS-NHS with amino groups at pag. 9.

- "NHS-NH₂ coupling reaction" should be written as "-NHS / NH₂-"

- Considering the separate activities of SCNPs@PTX and the SCNPs + laser (figure 5) it is quite surprising the strong anticancer activity of the combined therapy SCNP@PTX + laser, the same for the camptothecin case. The author should discuss more about these surprising results.

- Scheme 1 is interesting but a little bit too much condensed and it miss logic sequence of modifications.

In conclusion the paper needs major revisions.

We thank you and the two reviewers for the careful review of our article. We have revised the main text and the supporting information based on the comments and suggestions received.

Point-by-point response to the review comments:

1. *Reply to the first comment made by Referee 1 “The extent of cross-linking of the NHS-SS-NHS should be studied. Indeed, the length of the NHS-SS-NHS appears to be short for intermacromolecular bridging, as the distance between the amines in cyclodextrins appears be larger than the length of NHS-SS-NHS. Moreover, the possibility for binding with the cyclodextrin in the same polymer unit should be demonstrated. Finally, the residual NHS or COOH groups after the cross-linking should be quantified.”*

The corresponding corrections have been made. The extent of cross-linking was studied by determining the residual amine groups in the **SCNPs** using ninhydrin reaction (Scheme S2). As shown in Fig. S16, most of the amine groups (~84.7%) were consumed by the crosslinkers through –NHS/NH₂– reaction. The residual amine groups in **SCNPs** could be further used to conjugate NHS-DOTA and NHS-Cy5.5 for PET and fluorescence imaging. In the nanoparticles self-assembled from the polyrotaxanes, multiple H-bonds can be formed between the intermacromolecular β -CD-NH₂ (*Adv. Funct. Mater.* **2010**, *20*, 579). Thus, the distance between the intermacromolecular β -CD-NH₂ was suitable for NHS-SS-NHS to form intermacromolecular bridges. Although the reaction between NHS-SS-NHS and β -CD-NH₂ in the same polyrotaxane could not be completely avoided, most β -CD-NH₂ formed intermacromolecular bridges, because β -CD-NH₂ are mainly arranged in “tail-tail” mode due to the favorable hydrogen bonding. The distance between the amines on β -CD-NH₂ in “tail-tail” mode is much longer than that of the NHS-SS-NHS, thus significantly inhibiting the intramolecular crosslinking. To further demonstrate the successful formation of intermacromolecular bridges, we prepared seven polyrotaxanes with various β -CD-NH₂ and crosslinking efficiencies (Table S1 and Fig. S17). Fig. S18 indicates that the burst release of the loaded drugs was effectively inhibited by increasing the number of β -CD-NH₂ in the polyrotaxanes and their crosslinking efficiency, which further suggested that most of β -CD-NH₂ formed intermacromolecular bridges linked by the disulfide group. LC-MS was employed to measure the amount of residual NHS or COOH groups in **SCNPs** after crosslinking reaction. After reacting the solution of **SCNPs** with excess of tris(2-carboxyethyl)phosphine hydrochloride (TCEP) for 4 h, the small molecular weighted components cleaved from polyrotaxane were collected through ultrafiltration centrifugation (MWCO 10 kDa) for LC-MS study. No peaks were found corresponding to SHCH₂CH₂COOH (m/z = 106) or SHCH₂CH₂COONHS (sodium form m/z = 305, acid form m/z = 283), demonstrating the presence of negligible amount of NHS or COOH groups in **SCNPs**. The corresponding data and discussion have been added in the Supporting Information.

2. *Reply to the second comment made by Referee 1 “The stability of the particles in blood should be evaluated. From the PET results, the nanoparticles are showing extensive accumulation in liver and spleen, which may suggest that they are not functioning as designed. Particularly, I am worried about the aggregation and opsonization of the particles in blood.”*

The corresponding corrections have been made. We studied the stability of **SCNPs** in PBS containing 10% fetal bovine serum (FBS) at 37 °C by DLS. As shown in Fig. S26, negligible changes in size were detected during 24 h of incubation, confirming high colloidal stability of **SCNPs** in biological buffer. Additionally, **SCNPs@PTX** did not give rise to apparent hemolysis in the test concentration range (Fig. 3c), and the hemolysis percentage rose only slightly (1.1–4.3%) as concentration increased to the range of 50–200 µg PTX/mL, further demonstrating that **SCNPs** were stable enough in blood. The reason was that the hydrophilic segment (PEG) can form “brushlike” superstructures to prevent proteins from penetrating the surface and circumvent secondary adsorption onto the **SCNPs**. Like many other NPs, high accumulation of ⁶⁴Cu **SCNPs@DOTA** in liver and spleen was observed, because the ⁶⁴Cu **SCNPs@DOTA** were captured by the RES, which is a normal phenomenon for almost all nanomedicines. The data and the corresponding discussion have been added.

3. *Reply to the third comment made by Referee 1 “What is the effect of the rotaxanes in the polymers on drug release and stability? Studying various rotaxane amounts in the polymers will clarify this point.”*

The corresponding corrections have been made. We prepared several polyrotaxanes containing different numbers of β -CD-NH₂ by changing the ratio between **PDI-PCL-*b*-PEG-Mal** and β -CD-NH₂ during the preparation (Table S1). Drug release behaviors of the PTX-loaded **SCNPs** prepared from these polyrotaxanes were also studied. Fig. S18 indicated that the wheels (β -CD-NH₂) of the polyrotaxane could be crosslinked by **NHS-SS-NHS**, effectively inhibiting premature burst release of the loaded drugs. The data and the corresponding discussion have been added in the Supporting Information.

4. *Reply to the fourth comment made by Referee 1 “Besides the drug release, the disassembly of the particles at various GSH conditions should be evaluated. Particularly, I am interested to see if the photothermal induced cellular damage provokes an acceleration of the drug release and particle disruption.”*

The corresponding corrections have been made. We used TEM to study the morphology changes of the **SCNPs** in the presence of GSH with different concentrations. As shown in Fig. S28, no obvious size/morphology change of **SCNPs** was observed after incubating them with GSH for 24 h, because the structure of **SCNPs** could be maintained by strong π - π stacking between PDI groups and hydrophobic interactions between the PCL segments. Upon laser irradiation (671 nm, 0.5 W/cm²) for 10 min, **SCNPs** disassembled into an irregular aggregated form with larger diameter due to the photothermal effect (Fig. S28c). GSH was used to open the “molecular gates” of **SCNPs** through the cleavage of –SS– bonds, while laser irradiation was used to accelerate the release of the loaded drug through photothermal effect. Thus, the release rate was significantly increased upon laser irradiation in the presence of 10.0 mM GSH (Fig. 2i and Fig. S62). The data and the corresponding discussion have been added.

5. *Reply to the fifth comment made by Referee 1 “The PET imaging results on the accumulation of*

the particles in tumors should be correlated with the amount of paclitaxel being delivered to the tumor to that the particles avoid the premature drug release in vivo.”

The corresponding correction has been made. The biodistribution of PTX in the main organs was measured by HPLC, which was in good agreement with the result obtained from PET investigation (Fig. S35). The corresponding data and discussion have been added in the Supporting Information.

- 6. Reply to the sixth comment made by Referee 1 “While the authors used large tumors for PET imaging, the starting tumor size in the antitumor activity experiment is too small. Because a major issue for photothermal therapies is the poor light penetration into deep regions, the efficacy of the nanoparticles should also be studied in larger tumor masses to assess the potential of the system. Moreover, the authors are using subcutaneous HeLa xenografts for the evaluation of the antitumor effects. Such tumor model is not appropriate. Study the nanoparticles in orthotopic or metastatic tumors in mice with competent immune systems is more significant.”*

The corresponding corrections have been made. We also tested the anti-tumour performance of **SCNPs@PTX** on mice bearing larger tumours (around 220 mm³). As shown in Fig. S44 and Fig. S45, **SCNPs@PTX**-mediated chemotherapy and laser-irradiation-active PTT completely ablated the tumours without recurrence during the experimental period after a single-dose injection. The reason for this was that the combination of PTT and chemotherapy can effectively suppress the tumour growth. Moreover, a higher temperature activated the “switch” of **SCNPs@PTX** to release the loaded drugs, which diffused into a greater effective area, resulting in continuous chemotherapy effects, thus inhibiting tumour recurrence.

Furthermore, we studied the anti-tumour effect against orthotopic and metastatic tumour models. As shown in Fig. 6, Fig. S46–60, and Fig. S74–88, the supramolecular nanomedicines (**SCNPs@PTX** and **SCNPs@CPT**) exhibited superior anti-tumour and antimetastasis effect, as evidenced by the nearly complete ablation of the primary orthotopic tumours and effective prevention of metastasis to the lung, attributed to the synergy of PTT and chemotherapy. The data and detailed discussion have been added into the main text and Supporting Information.

- 7. Reply to the seventh comment made by Referee 1 “The novelty of the study should be clarified. Based on the authors explanations, the main novelty of the study is avoiding premature drug release from the carriers. However, there are already several formulations that can avoid premature drug release and control the delivery of drugs. From current manuscript, my impression is that the authors are just putting together various strategies, such as host-guest assembly with rotaxanes [Li, J., et al. Adv. Drug Deliv. Rev., 60 (2008) 1000-1017; etc.], combination of chemotherapy and photothermal therapy [Zou, L, et al. Theranostics 6 (2016) 762–772; etc.], active targeting by using cRGD peptide [Park, J. H., et al. J. Control. Release, 95 (2004) 579-588; etc.], into a complicated design.”*

Compared with traditional DDSs fabricated from conventional polymers, our polyrotaxane exhibited unique properties and advantages. The structure of this mechanically interlocked

molecule is uniquely designed with the stoppers, axles, and wheels all possessing special functions. In this structure, the targeting ligand, therapeutic, and imaging agents were perfectly integrated into one platform. For example, the RGD ligand acting as one of the stoppers endowed the resultant **SCNPs** with excellent targeting ability. Photothermal therapy (PTT) and photoacoustic imaging capabilities were introduced into our supramolecular theranostics by using PDI as the other stopper. The wheels (β -CD-NH₂) of the polyrotaxane served as “molecular gates” to avoid premature burst release and “reaction sites” to conjugate fluorescent dye (Cy5.5)/radiotracer (⁶⁴Cu DOTA) for fluorescence/PET imaging. On account of its dynamic nature, **SCNPs** prepared from the polyrotaxane acted as a “nanosponge”, with the ability to stably encapsulate a large amount of hydrophobic drugs, which would not be realized by traditional polymeric biomaterials. Due to the smart design, the supramolecular nanomedicines (**SCNPs@PTX** and **SCNPs@CPT**) completely ablated the tumours without recurrence after a single dose injection, benefiting from the combination of PTT and chemotherapy. Excitingly, these supramolecular nanomedicines also allowed treatment of highly aggressive orthotopic breast cancer and inhibited lung metastasis. Most importantly, the supramolecular nanomedicines constructed from the polyrotaxane not only showed fantastic topological structure, but also exhibited outstanding *in vivo* anti-tumour performances against different tumour models. To the best of our knowledge, this is the first report on the application of mechanically interlocked molecules in cancer theranostics. We firmly believe that this work will be a milestone in the application of supramolecular strategies in cancer theranostics, showing their unsurpassable advantages.

8. *Reply to the eighth comment made by Referee 1 “Instead of using I, II, III, IV,... in the figures, it would be much clear to use the names of the formulations.”*

The corresponding corrections have been made.

9. *Reply to the first comment made by Referee 2 “The syntheses of PDI-PCL, PDI-PCL-b-PEG, PDI-PCL-b-RGD=beta-CD-NH2 are briefly described in the “results and discussion” section but are missed in the “material and methods” section. So no details are given about these relevant procedures.”*

The detailed synthetic procedures are in the Supporting Information.

10. *Reply to the second comment made by Referee 2 “Some figures are wrongly cited in the body text, i.e. pag 5 Fig. 1 should be scheme 1, pag. 6 Fig. 2a should be Fig.1a, pag. 9 Fig. 3c should be Fig. 2c, etc. The Figures are sometimes not incrementally cited in the text, there are many supplemental figures not cited in the text and at least it is necessary to mention what can be found in the supplementary file.”*

The corresponding corrections have been made.

11. *Reply to the third comment made by Referee 2 “There are not experimental conditions on how the samples have been analysed by GPC, solvent, column and especially the standards used for*

column calibration.”

The corresponding corrections have been made. Detailed experimental conditions have been added in the Supporting Information.

12. *Reply to the fourth comment made by Referee 2 “GPC cannot be used to estimate the number of cyclodextrin included in PDI-PCL-b-RGD=beta-CD-NH2 because the MW estimation is accurate only when the standards are similar to the analysed sample. Here the standards are not specified but I don’t think a series of PDI-PCL-b-RGD=beta-CD-NH2 with different cyclodextrin units have been used. GPC separation is based on hydrodynamic volume and this might change dramatically with the addition of even a couple of cyclodextrins. So the number of seven cyclodextrins is not experimental validated.”*

The corresponding corrections have been made. We also utilized ^1H NMR spectroscopy to measure the number of $\beta\text{-CD-NH}_2$ in the polyrotaxane (Fig. S9, Fig. S17, and Table S1), which was calculated to be 7.3 by comparing the integrations of the peaks corresponding to the protons on $\beta\text{-CD-NH}_2$ with those of the PEG segment.

13. *Reply to the fifth comment made by Referee 2 “Regarding the preparation of polyrotaxanes, it is very important to demonstrate that the cyclodextrin is really pierced by PCL segment instead of having a conformation in which PCL forms a hairpin in the cavity of cyclodextrin. Also the last conformation will be stable and give similar results in the experiments used to characterize the polyrotaxane. Eventually only organic solvents may destabilize such conformation. Furthermore it has to be considered that to reach the PCL segment the cyclodextrin has to be pierced firstly by hydrophilic and flexible PEG starting from the maleimide group, a sequence of events which seems not so favoured in comparison to direct interaction with PCL hairpins.”*

The corresponding corrections have been made. We used various characterizations to confirm the successful preparation of polyrotaxane, such as ^1H NMR, 2D NOESY, FTIR, XRD, DSC, TGA and GPC studies, which provided convincing evidences for the formation of polyrotaxane. For example, DMSO is a good solvent that can be used to dissociate the host–guest complex if the PCL chain forms a hairpin in the cavity of $\beta\text{-CD-NH}_2$. In 2D NOESY spectrum, the NOE corrections were observed between the peaks corresponding to the protons on $\beta\text{-CD-NH}_2$ and the central protons of the PCL section (Fig. 1b), confirming that the PCL chain deeply penetrated into the cavity of $\beta\text{-CD-NH}_2$. As shown in GPC curves (Fig. 1a), the average molecular weight (M_n) of **PDI-PCL-b-PEG-RGD \supset β -CD-NH $_2$** was determined to be 16.8 kDa, which is 8.15 kDa higher than that of **PDI-PCL-b-PEG-RGD** ($M_n = 8.65$ kDa), providing direct evidence that polyrotaxane contains around seven $\beta\text{-CD-NH}_2$. In GPC study, DMF was utilized as an eluent, which can also dissociate the host–guest complex if the PCL chain forms a hairpin in the cavity of $\beta\text{-CD-NH}_2$. Moreover, FTIR, XRD, DSC, and TGA investigations also provided convincing evidences for the formation of polyrotaxane, which have been discussed in detail in the main text.

14. *Reply to the sixth comment made by Referee 2 “It is know that maleimide reacts with amines and*

when thiol groups are not present such reaction proceed quite well. Since I suppose PEG-PCL-b-PEG-MAL have been incubated with an excess of amino-cyclodextrin, I am wondering how such reaction has been prevented. The presence of amino groups should be verified by specific assay.”

The corresponding corrections have been made. Compared with the reaction between maleimide and thiol groups, the Michael addition reaction activity between maleimide and amine is much lower at room temperature in aqueous solution. The reaction between maleimide and amine could be effectively avoided in our synthetic conditions, which was confirmed by various characterizations. For example, the existence of cRGDfK in the polyrotaxane (**PDI-PCL-b-PEG-RGD**→**β-CD-NH₂**) was demonstrated by CLSM images (Fig. 3b, Fig. S29, and Fig. S65), cellular uptake studies (Fig. 3f and Fig. S64), and MTT assay (Fig. 3g). Moreover, we employed ¹H NMR spectroscopy to further demonstrate that the Michael addition between maleimide and amine was negligible during the synthesis of polyrotaxane. **PDI-PCL-b-PEG-Mal** (65.9 mg) and **β-CD-NH₂** (235 mg) were dissolved in the mixture of THF and water (1/1, v/v). The organic solvent was evaporated by stirring the solution for 12 h at room temperature. The solution was dialyzed against the mixture of DMSO/water (1/1, v/v) for 1 day (MWCO 12 kDa), then dialyzed against water for 1 day (MWCO 12 kDa) to eliminate free **β-CD-NH₂**. The aqueous solution was lyophilized for ¹H NMR study. As shown in Fig. S15b, negligible signals related to the protons of **β-CD-NH₂** were monitored, demonstrating that the amine group did not interfere with the reaction between maleimide and thiol groups. The presence of amino groups in the polyrotaxane was further verified by ninhydrin titration experiment (Fig. S16). The corresponding data and discussion have been added into the Supporting Information.

15. *Reply to the seventh comment made by Referee 2 “NHS-SS-NHS can also reacts with hydroxyl groups of cyclodextrins and also this reaction will lead to a decrease of zeta potential because part of the NHS-SS-NHS molecules may bind an hydroxyl group at one side and the other NHS will be hydrolysed to COOH by water instead to react with another OH.”*

The corresponding correction has been made. To demonstrate that **NHS-SS-NHS** reacted with the amine group rather than the hydroxyl groups on **β-CD-NH₂**, we utilized **β-CD** as a model compound. After stirring the mixture of **β-CD** (2.00 mM) and **NHS-SS-NHS** (1.00 mM) for 24 h in the presence of ethylenediamine (2.00 mM), the solution was dialyzed against water for 2 days (MWCO 1 kDa) and then lyophilized for ¹H NMR study. As shown in Fig. S14b, no signals were found that correspond to the methylene protons on **NHS-SS-NHS**, demonstrating that **NHS-SS-NHS** prefers to react with the amine group. Moreover, we tested the reaction between **NHS-SS-NHS** and hydroxyl groups in the absence of amine group. After stirring the mixture of **β-CD** (2.00 mM) and **NHS-SS-NHS** (1.00 mM) for 24 h, the solution was dialyzed against water for 2 days (MWCO 1 kDa) and lyophilized for ¹H NMR study. Similarly, no new signals were monitored (Fig. S14c), confirming that the reaction between **NHS-SS-NHS** and hydroxyl groups under the reaction condition was negligible. These results demonstrated that the hydroxyl groups on **β-CD-NH₂** did not interfere with the crosslinking reaction during the preparation of **SCNPs**. The corresponding data and discussion have been added into the Supporting Information.

16. *Reply to the eighth comment made by Referee 2 “NHS-SS-NHS is not anionic, please correct and adjust the discussion about electrostatic interaction of NHS-SS-NHS with amino groups at pag. 9.”*

As shown in Scheme 1, we utilized anionic **NHS-SS-NHS** containing sulfonate groups as the crosslinker.

17. *Reply to the ninth comment made by Referee 2 ““NHS-NH₂ coupling reaction” should be written as “-NHS / NH₂-””*

The corresponding correction has been made.

18. *Reply to the tenth comment made by Referee 2 “Considering the separate activities of SCNPs@PTX and the SCNPs + laser (figure 5) it is quite surprising the strong anticancer activity of the combined therapy SCNP@PTX + laser, the same for the camptothecin case. The author should discuss more about these surprising results.”*

The corresponding corrections have been made. The reasons behind the superior anti-tumour efficacy have been discussed in the main text. By taking full advantage of nanotechnology, high tumour accumulation of drug-loaded **SCNPs** was realized through both passive targeting (EPR effect) and active targeting. As one stopper of the polyrotaxane, the PDI group is an excellent photothermal agent, which can be employed in photothermal therapy (PTT). Followed by photothermal ablation of the main tumour tissue upon laser irradiation, the preloaded drugs can be released and diffused throughout the tumour region by photothermal-induced disruption of the NPs and enhancement of vascular permeability, effectively inhibiting tumour recurrence and avoiding the requirement for multiple doses. Moreover, the drug loading content and maximum tolerated dose of the supramolecular nanomedicines can be significantly improved, attributing to the unique topological structure of the polyrotaxane. Therefore, the therapeutic efficacy can also be enhanced by increasing the injection dose of anticancer drugs. As a result, superior anti-tumour performance can be achieved by combining PTT and chemotherapy.

19. *Reply to the eleventh comment made by Referee 2 “Scheme 1 is interesting but a little bit too much condensed and it miss logic sequence of modifications.”*

The corresponding correction has been made. We re-drew the cartoon shown in Scheme 1.

We highly appreciate all the valuable corrections suggested by the referees. We are hopeful that this revised manuscript now meets their standards for acceptance. Many thanks.

REVIEWERS' COMMENTS:

Reviewer #1 (Remarks to the Author):

The authors have successfully addressed most of my concerns. I believe the manuscript is ready for publication.

Reviewer #2 (Remarks to the Author):

In the response to reviewers the authors addressed the comments.

I could not find the MW of PEG, from page S15 lane 360 I can suppose that it is 2 kDa. I think that such short PEG forming a polyrotaxane by piercing about 7 molecules of beta-cyclodextrin will difficulty for a PEG brush over the NPs, helping to prevent protein adsorption on the surface of the NPs, maybe this fact can explain the liver accumulation of such NPs.

Point-by-point responses to the review comments:

1. *Reply to the first comment made by Referee 1 “The authors have successfully addressed most of my concerns. I believe the manuscript is ready for publication.”*

Thanks.

2. *Reply to the first comment made by Referee 2 “I could not find the MW of PEG, from page S15 lane 360 I can suppose that it is 2 kDa. I think that such short PEG forming a polyrotaxane by piercing about 7 molecules of beta-cyclodextrin will difficulty for a PEG brush over the NPs, helping to prevent protein adsorption on the surface of the NPs, maybe this fact can explain the liver accumulation of such NPs.”*

Thanks for the valuable suggestion. The molecular weight of PEG segment is 2 kDa, we have added this information in the “Supplementary Methods”.